# Single-cell multiomic profiling of human lungs reveals cell-type-specific and age-dynamic control of SARS-CoV2 host genes

Allen Wang[1†]*, Joshua Chiou[2,3†], Olivier B Poirion[1†], Justin Buchanan[1†], Michael J Valdez[2,3†], Jamie M Verheyden[3], Xiaomeng Hou[1], Parul Kudtarkar[3], Sharvari Narendra[3], Jacklyn M Newsome[3], Minzhe Guo[4,5], Dina A Faddah[6], Kai Zhang[7], Randee E Young[3,8], Justinn Barr[3], Eniko Sajti[3], Ravi Misra[9], Heidie Huyck[9], Lisa Rogers[9], Cory Poole[9], Jeffery A Whitsett[4,5], Gloria Pryhuber[9], Yan Xu[4,5], Kyle J Gaulton[3]*, Sebastian Preissl[1]*, Xin Sun[3,10]*, NHLBI LungMap Consortium[11]

[1]Center for Epigenomics & Department of Cellular & Molecular Medicine, University of California, San Diego, San Diego, United States; [2]Biomedical Sciences Graduate Program, University of California San Diego, La Jolla, United States; [3]Department of Pediatrics, University of California-San Diego, La Jolla, United States; [4]Division of Neonatology, Perinatal and Pulmonary Biology, Cincinnati Children's Hospital Medical Center, Cincinnati, United States; [5]Divisions of Pulmonary Biology and Biomedical Informatics, University of Cincinnati College of Medicine, Cincinnati, United States; [6]Vertex Pharmaceuticals, San Diego, United States; [7]Ludwig Institute for Cancer Research, La Jolla, United States; [8]Laboratory of Genetics, Department of Medical Genetics, University of Wisconsin-Madison, Madison, United States; [9]Department of Pediatrics and Clinical & Translational Science Institute, University of Rochester Medical Center, Rochester, United States; [10]Department of Biological Sciences, University of California-San Diego, La Jolla, United States; [11]NIH, Bethesda, United States

*For correspondence:
a5wang@health.ucsd.edu (AW);
kgaulton@health.ucsd.edu (KJG);
spreissl@health.ucsd.edu (SP);
xinsun@health.ucsd.edu (XS)

†These authors contributed equally to this work

**Abstract** Respiratory failure associated with COVID-19 has placed focus on the lungs. Here, we present single-nucleus accessible chromatin profiles of 90,980 nuclei and matched single-nucleus transcriptomes of 46,500 nuclei in non-diseased lungs from donors of ~30 weeks gestation,~3 years and ~30 years. We mapped candidate *cis*-regulatory elements (cCREs) and linked them to putative target genes. We identified distal cCREs with age-increased activity linked to SARS-CoV-2 host entry gene *TMPRSS2* in alveolar type 2 cells, which had immune regulatory signatures and harbored variants associated with respiratory traits. At the 3p21.31 COVID-19 risk locus, a candidate variant overlapped a distal cCRE linked to *SLC6A20*, a gene expressed in alveolar cells and with known functional association with the SARS-CoV-2 receptor ACE2. Our findings provide insight into regulatory logic underlying genes implicated in COVID-19 in individual lung cell types across age. More broadly, these datasets will facilitate interpretation of risk loci for lung diseases.

## Introduction

Amidst the ongoing COVID-19 pandemic, understanding how SARS-CoV-2 infects and impacts the lungs has become an urgent priority. Not only do the lungs act as a critical barrier that protects against inhaled pathogens such as viruses, it is also a site of many COVID-19 symptoms including the primary cause of COVID-19 mortality, acute respiratory distress syndrome (ARDS). The lungs are

composed of an elaborate airway tree that conducts air to and from the alveoli, the gas-exchange units. In an average human adult lungs, an estimated 480 million alveoli give rise to approximately 140 m$^2$ of gas-exchange surface area (*Ochs et al., 2004*). Airway and alveolar epithelium constitute the respiratory barrier that is exposed to inhaled pathogens. Respiratory epithelial cells are at the frontline of infection, although some pathogens that have bypassed the barrier can infect other cell types. The human airway epithelium is composed of luminal cells and basal cells (*Tata and Rajagopal, 2017*). Luminal cells include club cells and goblet cells that moisturize the air and trap pathogens, as well as ciliated cells that sweep out inhaled particles. These luminal cells are underlined by basal cells, which serve as progenitors when luminal cells are lost after infection (*Hogan et al., 2014*; *Kim, 2017*). The alveolar epithelium is composed of alveolar type 1 cells (AT1s), which are flat and line the gas–blood interface to facilitate gas exchange; and alveolar type 2 cells (AT2s), which produce surfactant to reduce surface tension and protect against pathogens (*Whitsett and Weaver, 2015*). While SARS-CoV-2 likely infects both the airway and alveolar regions of the lungs, it is the damage to the alveolar region that causes ARDS (*Du et al., 2020*).

There are several large-scale studies, including efforts from LungMap and the Human Cell Atlas, which aim to define cell types within the human lungs using single-cell transcriptomics as the central modality (*Reyfman et al., 2019*; *Schiller et al., 2019*; *Travaglini et al., 2020*; *Xu et al., 2016*). In contrast, there is a paucity of single-cell data focused on mapping *cis*-regulatory elements (CREs) in the human genome that are active in specific lung cell types. CREs associate with combinations of transcription factors to drive spatiotemporal patterns of gene expression (*Moore et al., 2020*) and enable cell-specific responses to intra- and extra-cellular signals, for example, aging (*Booth and Brunet, 2016*) and inflammation (*Smale and Natoli, 2014*). Furthermore, complex disease-associated variants identified in genome-wide association studies (GWAS) are enriched in CREs (*Maurano et al., 2015*; *Pickrell, 2014*). Therefore, a comprehensive atlas of cell-type resolved CREs in the human lungs will facilitate investigation of the gene regulatory mechanisms responsible for lung cell-type identity, function, and role in biological processes such as viral entry, as well as uncovering the effects of genetic variation on complex lung disease.

Accessible or 'open' chromatin is a hallmark of CREs and can be used to localize candidate *cis*-regulatory elements (cCREs). Chromatin accessibility can be assayed using 'bulk' or 'ensemble' techniques such as DNase-seq and ATAC-seq (*Buenrostro et al., 2013*; *Thurman et al., 2012*). To overcome limitations regarding tissue heterogeneity inherent in such assays, technologies such like single-cell ATAC-seq have been developed to map the epigenome and gene regulatory programs within component cell types (*Buenrostro et al., 2015*; *Chen et al., 2018*; *Cusanovich et al., 2015*; *Cusanovich et al., 2018*; *Lareau et al., 2019*; *Satpathy et al., 2019*). Accessible chromatin profiles derived from single cells can elucidate cell-type-specific cCREs, transcriptional regulators driving element activity, and putative target genes linked to distal cCREs through single-cell co-accessibility (*Cusanovich et al., 2018*; *Lareau et al., 2019*; *Pliner et al., 2018*; *Preissl et al., 2018*; *Satpathy et al., 2019*). Importantly, human sequence variants affecting complex traits and diseases are enriched in non-coding sequences (*Maurano et al., 2015*; *Pickrell, 2014*). Thus, cell-type-specific profiles derived from single-cell chromatin accessibility data can help prioritize the cell types of action and function of these variants (*Chiou et al., 2019*; *Corces et al., 2020*).

Epidemiology data of US cases reported by the CDC has consistently demonstrated that the rate of hospitalization or death from COVID-19 is significantly lower among children compared to adults or elderly individuals, amidst caution that children can still be infected and transmit the virus (*CDC, 2020a*; *CDC, 2020b*). There are likely many reasons that underlie the age-associated differences, including different expression levels of viral entry proteins and different immune resilience to viral infection. Defining the mechanism underlying the apparent reduced susceptibility of children to COVID-19 will inform how we can transfer this advantage to adult and elderly populations.

Both in silico structural modeling and biochemical assays have implicated several key host proteins for SARS-CoV-2 infection. ACE2 has been demonstrated as the receptor for not only the original SARS-CoV, but also SARS-CoV-2 (*Hoffmann et al., 2020*; *Lan et al., 2020*; *Yan et al., 2020*). Based on literature from the original SARS-CoV as well as emerging data from SARS-CoV-2, TMPRSS2 and CTSL cleave the viral spike protein, thereby facilitate fusion of the virus with host cells (*Huang et al., 2006*; *Matsuyama et al., 2020*; *Reinke et al., 2017*; *Walls et al., 2020*; *Zhou et al., 2016*). In particular, TMPRSS2 has been shown to be essential for coronavirus viral entry while CTSL is dispensable (*Hoffmann et al., 2020*; *Shirato et al., 2018*; *Zhou et al., 2015*). *BSG* encodes

another receptor that can bind to the SARS-CoV spike protein (*Chen et al., 2005*) and *FURIN* encodes a protease with a putative target site in SARS-CoV-2, adding both genes to the list of host machinery highjacked by the virus (*Coutard et al., 2020*; *Walls et al., 2020*). In this study, we focus on the genes encoding these five proteins, *ACE2*, *TMPRSS2*, *CTSL*, *BSG*, and *FURIN*, and determine their expression and associated *cis*-regulatory landscape at single-cell resolution in the non-diseased human lungs.

To contribute to our understanding of gene regulation in the human lungs during aging and how such regulation goes awry and contributes to disease, including SARS-CoV-2 infection, we generated donor-matched single-nucleus RNA-seq and single-nucleus ATAC-seq data across neonatal, pediatric, and adult lungs with three donors in each group. Using these datasets, we profiled gene expression dynamics at cell-type resolution of SARS-CoV-2 host entry genes *ACE2*, *TMPRSS2*, *CTSL*, *BSG*, and *FURIN* and revealed cCREs underlying these changes for *ACE2* and *TMPRSS2*, genes that encode the primary receptor and fusion protein. We further profiled non-coding sequence variation in cCREs associated with *TMPRSS2* that may impact regulatory activity and might contribute to differential susceptibility to SARS-CoV-2 infection by affecting *TMPRSS2* expression. Finally, we demonstrated the value of this resource in interpreting emerging genetic risk of respiratory failure in COVID-19 by annotating the recently identified 3p21.31 locus (*Ellinghaus et al., 2020*).

## Results

### Single-nucleus accessible chromatin and transcriptional profiles from neonatal, pediatric, and adult human lung tissues

To generate an age and cell-type resolved atlas of chromatin accessibility and gene expression in the human lungs, we performed single-nucleus ATAC-seq (snATAC-seq) and single-nucleus RNA-seq (snRNA-seq) on non-diseased lung tissue sourced from the NIH funded LungMap Human Tissue Core. Tissue samples spanned three donor age groups: ~30-week-old gestational age (GA, prematurely born, 30wk$^{GA}$), ~3-year-old (3yo), and ~30-year-old (30yo) (metadata in *Supplementary file 1*). After batch correction and filtering of low-quality nuclei and likely doublets, we clustered and analyzed a total of 90,980 single-nucleus accessible chromatin profiles (*Figure 1A*, and *Figure 1—figure supplement 1A–D*, *Supplementary file 2*). We identified 19 clusters representing epithelial (AT1-alveolar type 1, AT2-alveolar type 2, club, ciliated, basal, and pulmonary neuroendocrine), mesenchymal (myofibroblast, pericyte, matrix fibroblast 1, and matrix fibroblast 2), endothelial (arterial, lymphatic, capillary 1 and capillary 2), and hematopoietic cell types (macrophage, B-cell, T-cell, NK cell, and enucleated erythrocyte) (*Figure 1A*). Supporting these cluster annotations, we observed cell-type-specific patterns of chromatin accessibility at known marker genes for each cell type (*Figure 1B*, and *Figure 1—figure supplement 2A*). We similarly clustered the 46,500 single-nucleus transcriptomes, which passed QC criteria from the donor and sample-matched snRNA-seq data (*Figure 1C*, and *Figure 1—figure supplement 1E–H*, *Supplementary file 2*). These clusters represented all major cell types in the small airway region of the lungs (*Figure 1C,D*, and *Figure 1—figure supplement 2B*). Importantly, these clusters overlapped those identified from snATAC-seq, highlighted by a cluster of rare pulmonary neuroendocrine cells (PNECs) represented in both modalities (*Figure 1A–D*, *Figure 1—figure supplement 2A,B*).

### Cell-type-specific expression and regulation of SARS-CoV-2 host cell entry genes

To gain insight into how viral entry is regulated in host cell types, we set out to identify the CREs predicted to regulate SARS-CoV-2 cell entry factors and to pinpoint the cell types in which they exert their effects. Toward this goal, we first identified the discrete cell types that express *ACE2*, *TMPRSS2*, *CTSL*, *BSG*, and *FURIN*. We detected *ACE2* transcript in very few nuclei (total 80 nuclei) in the normal lungs and these nuclei were enriched within the epithelial lineage (*Figure 2A*, *Figure 2—figure supplement 1A*, *Supplementary file 3*). This is consistent with exceptionally low *ACE2* expression in multiple tissues analyzed in recent publications (*Muus et al., 2020*; *Qi et al., 2020*; *Sungnak et al., 2020*; *Zhao et al., 2020*; *Ziegler et al., 2020*; *Zou et al., 2020*). In our data, AT2 cells had the highest number of *ACE2*$^+$ nuclei, accounting for 48.8% of all *ACE2*-expressing nuclei (39 out of total 80 *ACE2*$^+$ nuclei) (*Figure 2—figure supplement 1A*, *Supplementary file 3*).

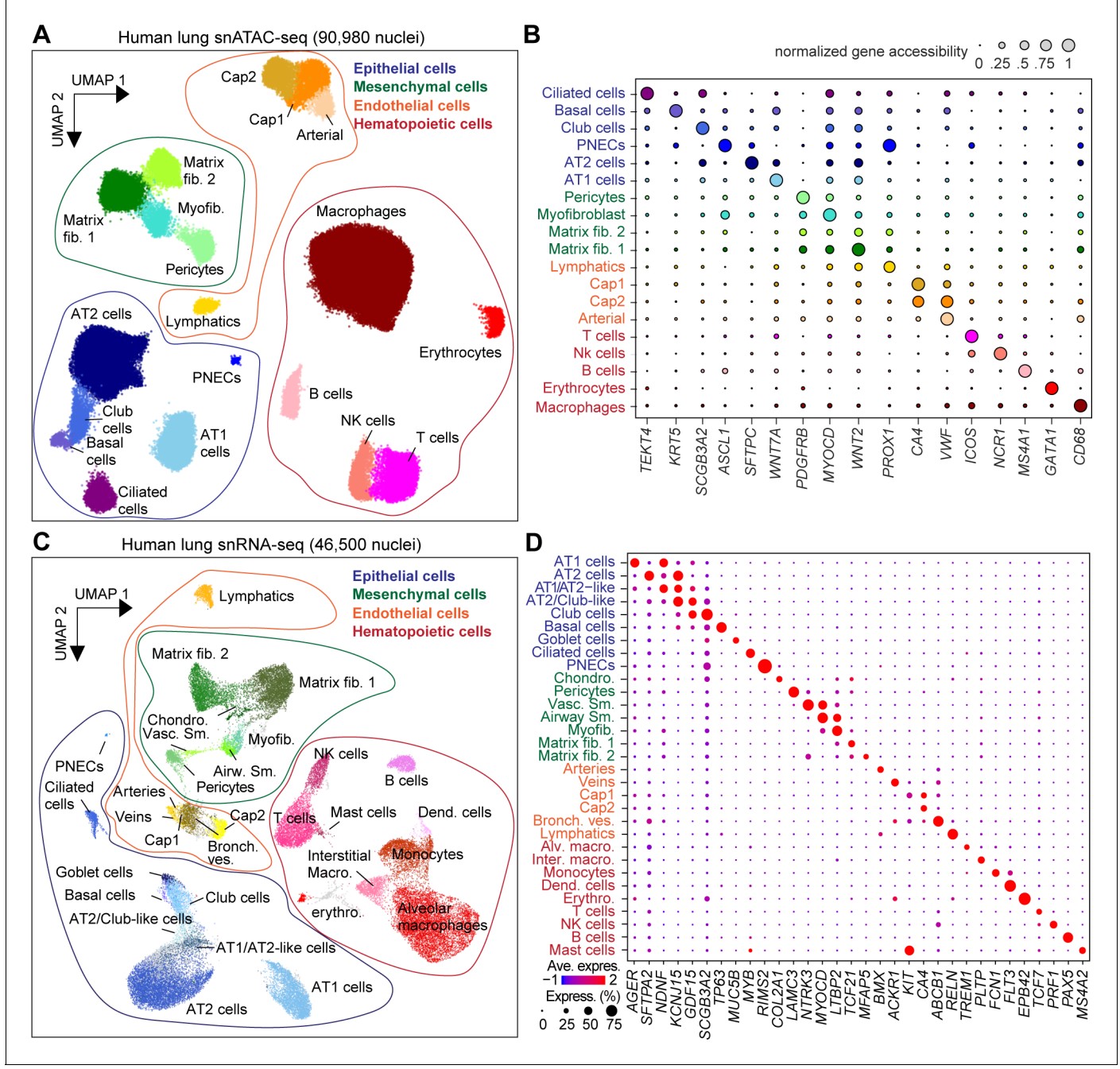

**Figure 1.** Single-nucleus atlas of chromatin accessibility and transcriptomes in the human lungs. (**A**) UMAP (Uniform Manifold Approximation and Projection) embedding (*McInnes et al., 2018*) and clustering results of snATAC-seq data from 90,980 single-nucleus chromatin profiles from ten donors: premature born (30 week[GA] for gestational age, n = 3), 4-month-old (n = 1), three yo (n = 3) and 30 yo (n = 3). For library quality control see *Figure 1—figure supplement 1A–D*. (**B**) Dot plot of marker genes from snRNA-seq used for cluster annotation. For additional genes see *Figure 1—figure supplement 2A*. (**C**) UMAP embedding (*McInnes et al., 2018*) and clustering result of 46,500 snRNA-seq data from nine donors: premature born (30 week[GA]), three yo, 30 yo, n = 3 per time point, identifies 31 clusters. Each dot represents a nucleus. Spread-out gray dots correspond to nuclei of unclassified cells. For library quality control see *Figure 1—figure supplement 1E–H*. (**D**) Dot plot of marker genes from snRNA-seq used for cluster annotation. For additional genes see *Figure 1—figure supplement 2B*.

The online version of this article includes the following figure supplement(s) for figure 1:

**Figure supplement 1.** Quality control of snATAC-seq and snRNA-seq datasets.

**Figure supplement 2.** Expression and chromatin accessibility at marker gene loci used for annotation.

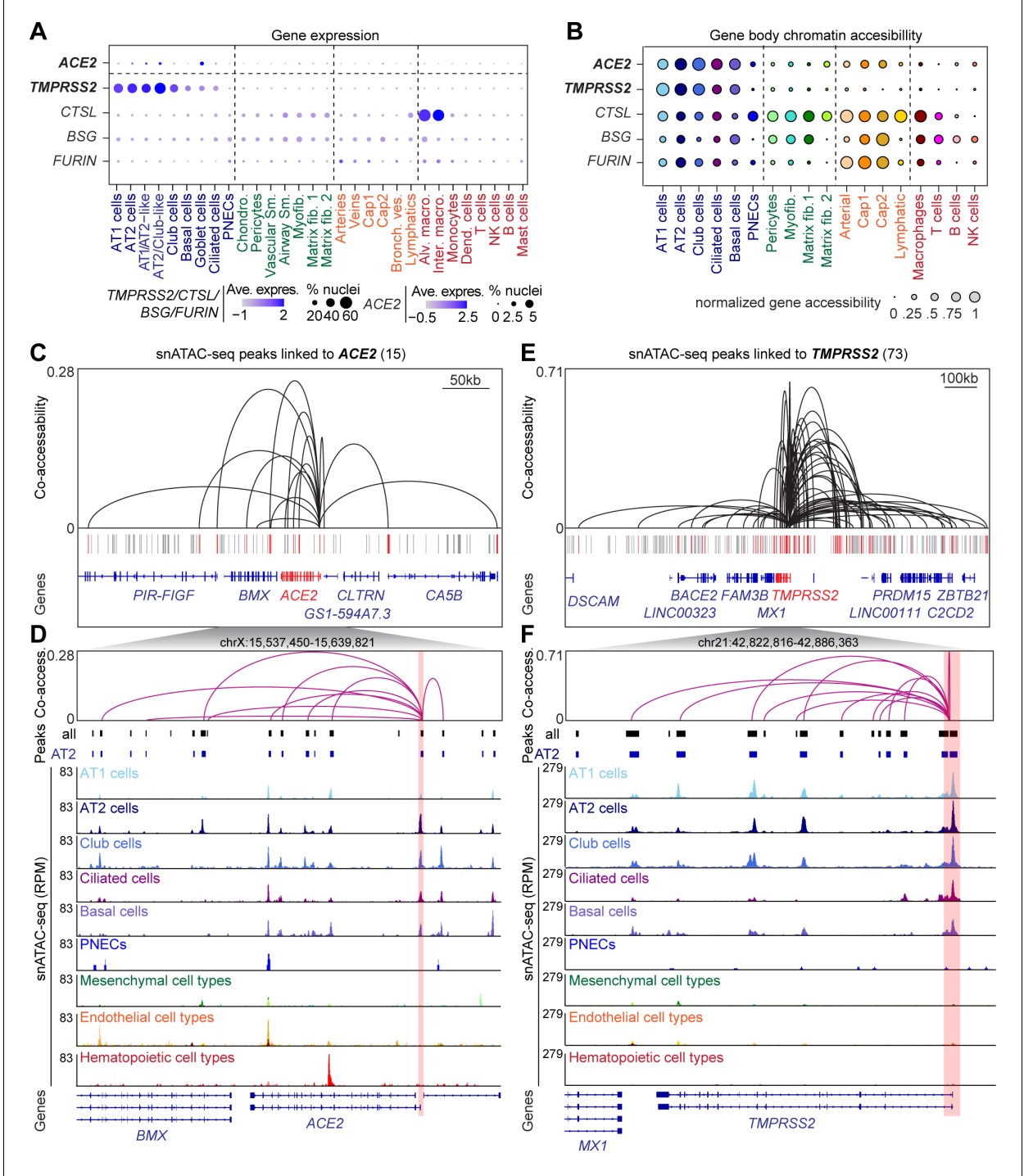

**Figure 2.** snATAC-seq analysis of human lungs reveals candidate *cis*-regulatory elements for *ACE2* and *TMPRSS2*. (**A**) Dot plot illustrating cluster-specific gene expression of candidate SARS-CoV-2 cell entry genes. For violin plots illustrating cluster-specific gene expression please see *Figure 2—figure supplement 1A–E*. (**B**) Dot plot illustrating cluster-specific gene body chromatin accessibility of candidate SARS-CoV-2 cell entry genes. (**C**) Union set of peaks (vertical lines) identified in all clusters surrounding *ACE2* and 15 peaks that showed co-accessibility with the *ACE2* promoter (red lines, co-accessibility score >0.05) via Cicero (*Cusanovich et al., 2018*). (**D**) Zoom into *ACE2* locus and genome browser tracks of snATAC-seq signal (*Robinson et al., 2011*). *ACE2* promoter region highlighted by red box. (**E**) Union set of peaks (vertical lines) identified in all clusters surrounding *TMPRSS2* and 73 peaks that showed co-accessibility with the *TMPRSS2* promoter (red lines, co-accessibility score >0.05) via Cicero (*Cusanovich et al., 2018*). (**F**) Zoom into *TMPRSS2* locus and genome browser tracks of snATAC-seq signal (*Robinson et al., 2011*). *TMPRSS2* promoter region highlighted by red box. For genome browser tracks of *BSG*, *FURIN*, *CTSL* please see *Figure 2—figure supplement 1F*.
*Figure 2 continued on next page*

*Figure 2 continued*

The online version of this article includes the following figure supplement(s) for figure 2:

**Figure supplement 1.** Gene expression and chromatin accessibility for SARS-COV-2 cell entry genes.

In comparison, *TMPRSS2* transcripts were detected in many more cells (total 6547 nuclei, *Figure 2A*, *Figure 2—figure supplement 1B*, *Supplementary file 3*). Most *TMPRSS2*-expressing cells were epithelial cells including AT1 and AT2 cells and airway cells such as club, ciliated and goblet cells (*Figure 2A*, *Figure 2—figure supplement 1B*, *Supplementary file 3*). Within the AT2 population, *TMPRSS2* was detected in 3,315/7,226 nuclei, or 45.8% of the AT2 cells (*Figure 2—figure supplement 1B*). Importantly, 21 of the 39 *ACE*+ AT2 cells also expressed *TMPRSS2* (*Supplementary file 3*). The other three candidate genes of SARS-CoV-2 host cell entry *CTSL*, *BSG* and *FURIN* were expressed in a large number of AT1, AT2, matrix fibroblast1,2, and M1 macrophage cells, as well as a small number of cells in additional cell types (*Figure 2A*, *Figure 2—figure supplement 1C–E*, *Supplementary file 3*).

We next assessed cell-type resolved chromatin accessibility at candidate SARS-CoV-2 entry genes. Consistent with their gene expression, both *ACE2* and *TMPRSS2* were primarily accessible throughout their gene body in alveolar cells such as AT1, AT2, and airway cells such as club, ciliated, and basal cells (*Figure 2B*). Conversely, the *CTSL* gene body exhibited chromatin accessibility across epithelial cells, mesenchymal cells, endothelial cells, and macrophages (*Figure 2B*, *Figure 2—figure supplement 1F*). *BSG* and *FURIN* also showed broad chromatin accessibility patterns with the highest activity in endothelial cells, such as capillaries (*Figure 2B*, *Figure 2—figure supplement 1F*). Together, both gene expression and chromatin accessibility suggest that among cell types constituting the barrier exposed to inhaled pathogens, both the airway and alveolar epithelial cells express genes critical for SARS-CoV-2 entry.

Cell-type-specific expression profiles are largely established by distal CREs such as enhancers (*ENCODE Project Consortium, 2012*; *Moore et al., 2020*; *Kundaje et al., 2015*). To identify cCREs predicted to control cell-type-restricted expression of the SARS-CoV-2 viral entry genes, we first aggregated nuclei within each cell type. We then called accessible chromatin sites from the aggregated profiles using MACS2 (*Zhang et al., 2008*). Overall, we mapped 398,385 cCREs across all lung cell types. Distal cCREs can be linked to putative target genes by measuring co-accessibility with promoter regions, as it has been shown that co-accessible sites tend to be in physical proximity in the nucleus (*Pliner et al., 2018*). As such, we identified sites co-accessible with the *ACE2*, *TMPRSS2*, *CTSL*, *FURIN*, and *BSG* promoters using a modified implementation of Cicero (*Pliner et al., 2018*). At the *ACE2* locus, we identified 15 sites co-accessible with the *ACE2* promoter (*Figure 2C,D*, *Supplementary file 4*). We speculate that the modest number of co-accessible sites is likely due to the small percentage of *ACE2*+ nuclei (*Figure 2A*, *Figure 2—figure supplement 1A*). In comparison, at the *TMPRSS2* locus, we identified 73 accessible chromatin sites co-accessible with the *TMPRSS2* promoter (*Figure 2E,F*, *Supplementary file 4*). Finally, at the *CTSL*, *FURIN*, and *BSG* loci we identified 73, 213, and 64 accessible chromatin sites co-accessible with their respective gene promoters (*Supplementary file 4*). This collection of cell-type resolved cCREs associated with SARS-CoV-2 host genes (*Supplementary file 4*) will be crucially important for follow-up studies to determine how host cell genes are regulated and how genetic variation within these elements contributes to infection rate and disease outcomes.

## CREs linked to *TMPRSS2* are part of an age-related regulatory program in AT2 cells

AT2 cells are an abundant epithelial cell type in the alveolar region of the lungs where COVID-19 disrupts respiration. Consequently, we focused on AT2 cells to evaluate viral entry gene dynamics across donor age groups (*Figure 3*). We observed a higher fraction of AT2 cells expressing *ACE2* and *TMPRSS2* in adult lungs as compared to pediatric samples in our small cohort (n = 3 per age group, *Figure 3A,B*). Notably, these observed age-related increase in expression of these two genes is consistent with findings from a parallel report spearheaded by the Human Cell Atlas (HCA) that included pediatric data as part of a large-scale meta-analysis (*Muus et al., 2020*; *Schuler et al., 2020*). In contrast to the percentage of AT2 cells expressing these genes, the expression levels per

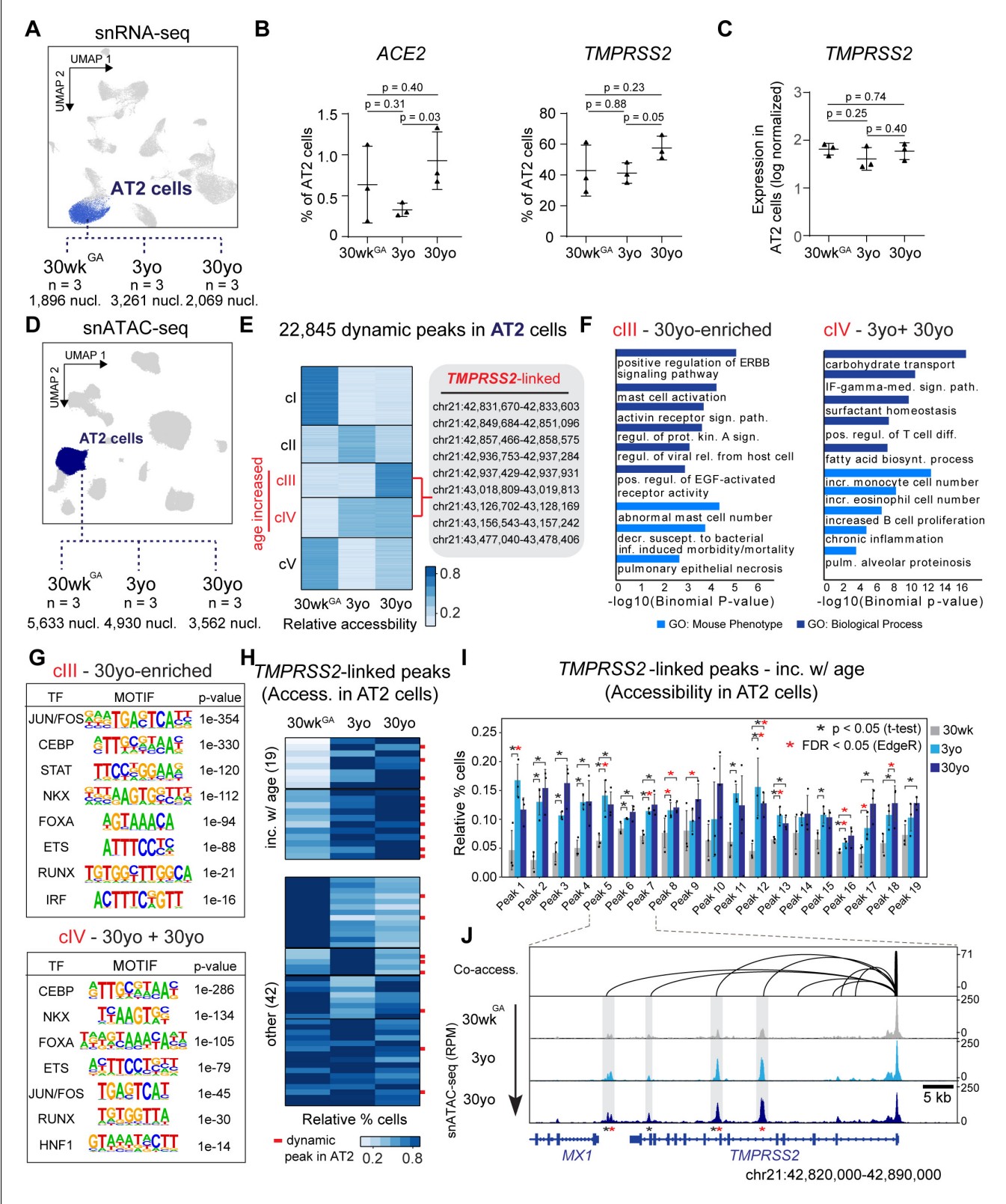

**Figure 3.** Age-increasing gene expression and accessible chromatin in AT2 cells exhibits signatures of immune regulation and harbors *TMPRSS2*-linked sites of chromatin accessibility. (**A**) Differential analysis was performed on AT2 cells between three ages with replicates (n = 3 per stage). (**B**) Fraction of AT2 cells with expression of *ACE2* (left) and *TMPRSS2* (right) in 30wk^GA, 3yo and 30yo human lung samples. All data are represented as mean ± SD. p values derived from unpaired, two-tailed t-tests. For expression data of *BSG, CTSL, FURIN* please see *Figure 3—figure supplement 1A*. (**C**) Log

*Figure 3 continued on next page*

*Figure 3 continued*

normalized expression of *TMPRSS2* in AT2 cells. Displayed are median expression values for AT2 cells in individual samples with at least 1 UMI (unique molecular identifier). (D) Differential analysis was performed on AT2 cells using pairwise comparisons between three ages with replicates (n = 3 per stage). (E) K-means cluster analysis (K = 5) of relative accessibility scores (see Materials and methods) for 22,845 age-dynamic peaks (FDR < 0.05, EdgeR) (*Robinson et al., 2010*) in AT2 cells. Clusters III and IV show increasing accessibility with age and contain nine *TMPRSS2*-co-accessible sites. (F) GREAT (*McLean et al., 2010*) analysis of elements in group cIII (left panel) and cIV (right panel) shows enrichment of immune-related gene ontology terms. (G) Transcription factor motif enrichment analysis of elements in cIII and cIV. (H) K-means cluster analysis (K = 6) of *TMPRSS2*-co-accessible sites based on the relative percentage of AT2 cells with at least one fragment overlapping each peak. Red bars indicate dynamic peaks identified from pairwise differential analysis (FDR < 0.05, EdgeR) (*Robinson et al., 2010*). (I) Locus restricted differential analysis of *TMPRSS2*-linked peaks with increased accessibility in AT2 with age (top panel in 3H). Data are represented as mean ± SD. Black asterisk, p<0.05 (independent t-test); Red asterisk, FDR < 0.05 (EdgeR) (*Robinson et al., 2010*) from dynamic peak analysis. For additional sites and promoter accessibility of *TMPRSS2* please see *Figure 3—figure supplement 1B,C*. (J) Genome browser representation of four *TMPRSS2*-linked peaks across age groups (*Robinson et al., 2011*).

The online version of this article includes the following source data and figure supplement(s) for figure 3:

**Source data 1.** Normalized expression values for *TMPRSS2* in AT2 cells.
**Figure supplement 1.** Gene expression of additional SARS-COV-2 cell entry genes and chrmatin accessibility of peak linked to *TMPRSS2* during aging.

nucleus were similar across different age groups for either *ACE2* (no nucleus had >1 UMI detected) or *TMPRSS2* (*Figure 3C*). Notably, we did not observe an age-related trend for the other candidate viral entry genes *BSG, CTSL, FURIN* (*Figure 3—figure supplement 1A*).

We next leveraged our snATAC-seq data to identify cCREs predicted to control cell-type-specific and age-related gene expression of SARS-CoV-2 cell entry genes. We focused on *TMPRSS2* as it is essential for coronavirus entry into host cells (*Hoffmann et al., 2020*; *Shirato et al., 2018*; *Zhou et al., 2015*). Compared to *ACE2*, *TMPRSS2* was detected in sufficient number of cells to allow us power to address its regulation. Having identified cCREs predicted to regulate *TMPRSS2* expression (*Figure 2E,F*), we speculated that some of these sites could modulate the age-associated increase of *TMPRSS2* expressing AT2 cells (*Figure 3B*). To examine this in an unbiased fashion, we first identified genome-wide chromatin sites in AT2 cells that show dynamic accessibility across donor age groups. We tested all possible pairwise age comparisons between AT2 signal from each of the three groups of 30wk[GA], 3yo, and 30yo donors while accounting for donor to donor variability (*Figure 3D*, see Materials and methods). Overall, we identified 22,745 age-linked sites in AT2 cells, which exhibited significant differences (FDR < 0.05) in any pairwise comparison (*Figure 3D,E*). Clustering of these dynamic peaks revealed five predominant groups of age-linked chromatin accessibility patterns (cI-cV, *Figure 3E*). Given the sample size limitation (n = 3 per age group), we acknowledge that the statistical significance of these observed dynamic changes will require further corroboration using datasets from additional donor samples. Nevertheless, we reasoned that because these changes are observable despite modest sample size, the trends provide informative biological insights.

Of these dynamic peaks, we identified two clusters of AT2 sites exhibiting increasing accessibility with age including several sites linked to candidate genes for SARS-CoV-2 host genes, most notably nine sites co-accessible with *TMPRSS2* (cIII 30yo enriched and cIV 3yo + 30yo, *Figure 3E*). Intriguingly, these two age-increasing co-accessible site containing clusters were enriched for processes related to viral infection, immune response and injury repair such as viral release from host cell, interferon-gamma mediated signaling pathway, and positive regulation of ERBB signaling pathway (*Figure 3F*, *Supplementary file 5*). Also, these age-dependent clusters were enriched for phenotypes substantiated in mouse studies, such as pulmonary epithelial necrosis, increased monocyte cell number, and chronic inflammation (*Figure 3F*, *Supplementary file 5*). We observed an enrichment of sequence motifs within these clusters for transcription factors controlling endoderm cell fate (FOXA, HNF1), lung cell fate (NKX), AT2 cell fate (CEBP) and AT2 cell signaling (ETS) (*Maeda et al., 2007*; *Morrisey et al., 2013*; *Morrisey and Hogan, 2010*). Further supporting immune regulation of AT2 cell gene expression, we observed an enrichment of motifs for factors involved in immune

signaling such as STAT, IRF, and FOS/JUN (*Au-Yeung and Horvath, 2018*; *Mogensen, 2018*; *Figure 3G*, *Supplementary file 6*).

To complement the genome-wide unbiased approach which identified 9 *TMPRSS2* co-accessible sites as age-increasing (*Figure 3E*), we next assessed in a locus restricted manner how many of the 73 co-accessible sites (*Figure 2D*) showed increased accessibility with age in AT2 cells. Overall, we identified 10 additional cCREs co-accessible with *TMPRSS2,* which exhibited patterns of increasing accessibility with age for a total of 19 age-increasing *TMPRSS2*-linked cCREs, 17 of which were statistically significant, with the caveat of modest sample size (N = 3 per age group) (FDR < 0.05 via EdgeR and/or p<0.05 via independent t-test, *Figure 3H,I*, *Figure 3—figure supplement 1C*, *Supplementary file 4*). When viewed in genomic context, several of these sites showed a clear age-linked increase in read depth likely reflecting a higher fraction of accessible nuclei (*Figure 3J*). Notably, accessibility at the *TMPRSS2* promoter did not exhibit differential accessibility with age (*Figure 3J*, *Figure 3—figure supplement 1B*) emphasizing a likely role of distal cCREs in regulating age-increasing *TMPRSS2* expression in AT2 cells.

## Genetic variants predicted to affect age-increased *TMPRSS2*-linked cCREs are associated with respiratory phenotypes and *TMPRSS2* expression

Mapping distal cCREs linked to *TMPRSS2* allowed us to next identify non-coding sequence variation that might affect *cis*-regulatory activity and contribute to inter-individual differences in *TMPRSS2* expression and the risk of lung disease. We therefore characterized genetic variation in the 19 cCREs with age-increased chromatin accessibility and linked to *TMPRSS2* in AT2s (*Figure 3H,I*).

In total, 2270 non-singleton sequence variants in the gnomAD v3 database (*Karczewski et al., 2019*) overlapped age-increasing cCREs linked to *TMPRSS2* in AT2s. To determine which of these variants might affect regulatory activity in AT2 cells, we first identified variants in predicted sequence motifs of transcription factor (TF) families such as CEBP, ETS, NKX, FOXA, IRF and STAT which were enriched in AT2 cCREs. In total we identified 1100 variants in a predicted motif for one or more of these TFs (*Figure 4A*, *Supplementary file 7*). We further applied a machine learning approach (deltaSVM) (*Lee et al., 2015*) to model AT2 chromatin accessibility and identified 212 variants with significant predicted effects (FDR < 0.1) on AT2 chromatin accessibility (*Figure 4A*, *Supplementary file 7*). Among motif-bound variants, 50 were common (defined here as minor allele frequency [MAF]>1%) of which 10 further had predicted effects on AT2 chromatin accessibility using deltaSVM (*Lee et al., 2015*; *Figure 4A*, *Supplementary file 7*). Common variants with predicted function generally had consistent frequencies across populations, although multiple variants, for example rs35074065, were much less common in East Asians (MAF = 0.005) relative to other populations (Europeans MAF = 0.45, South Asian MAF = 0.37, African MAF = 0.12).

We next determined whether common variants with predicted AT2 regulatory effects were associated with phenotypes related to respiratory function, infection, medication use or other traits using GWAS summary statistic data generated using the UK Biobank (UKBB) (*Sudlow et al., 2015*). Among the 10 common variants that were both TF motif-disrupting and had predicted effects on AT2 chromatin accessibility, the most significant association was between rs35074065 and emphysema (p=5.64 $\times$ 10$^{-7}$) (*Figure 4B*). This variant also had evidence for association with asthma (p=6.7 $\times$ 10$^{-4}$). Furthermore, the majority of these variants (9/10) were nominally associated (p<1$\times$10$^{-2}$) with at least one phenotype related to respiratory function or respiratory medication use including bronchiectasis (rs462903 p=2.0 $\times$ 10$^{-4}$, rs9974995 p=7.1 $\times$ 10$^{-4}$), bacterial pneumonia (rs2838089 p=2.4$\times$10$^{-4}$), COPD (rs1557372 p=2.9 $\times$ 10$^{-3}$), asthma (rs8127290 p=1.4$\times$10$^{-3}$) and medications used to treat asthma such as serevent (rs220266 p=3.1$\times$10$^{-4}$, rs62219349 p-5.3 $\times$ 10$^{-3}$) (*Figure 4B*).

Given that common AT2 variants showed predicted regulatory function and association with respiratory disease, we next asked whether these variants regulated the expression of *TMPRSS2* using human lung eQTL (expression quantitative trait loci) data from the GTEx v8 release (*GTEx Consortium, 2020*). Among variants tested for association in GTEx, we observed a highly significant eQTL for *TMPRSS2* expression at rs35074065 (p=3.9 $\times$ 10$^{-11}$) as well as more nominal eQTL evidence at rs1557372 (p=2.9 $\times$ 10$^{-5}$) and rs9974995 (p=3.5 $\times$ 10$^{-6}$). Furthermore, in fine-mapping data from GTEx, rs35074065 had a high posterior probability (PPA = 41.6%) and therefore likely has a direct casual effect on *TMPRSS2* expression (*Figure 4C*). This variant further disrupted predicted

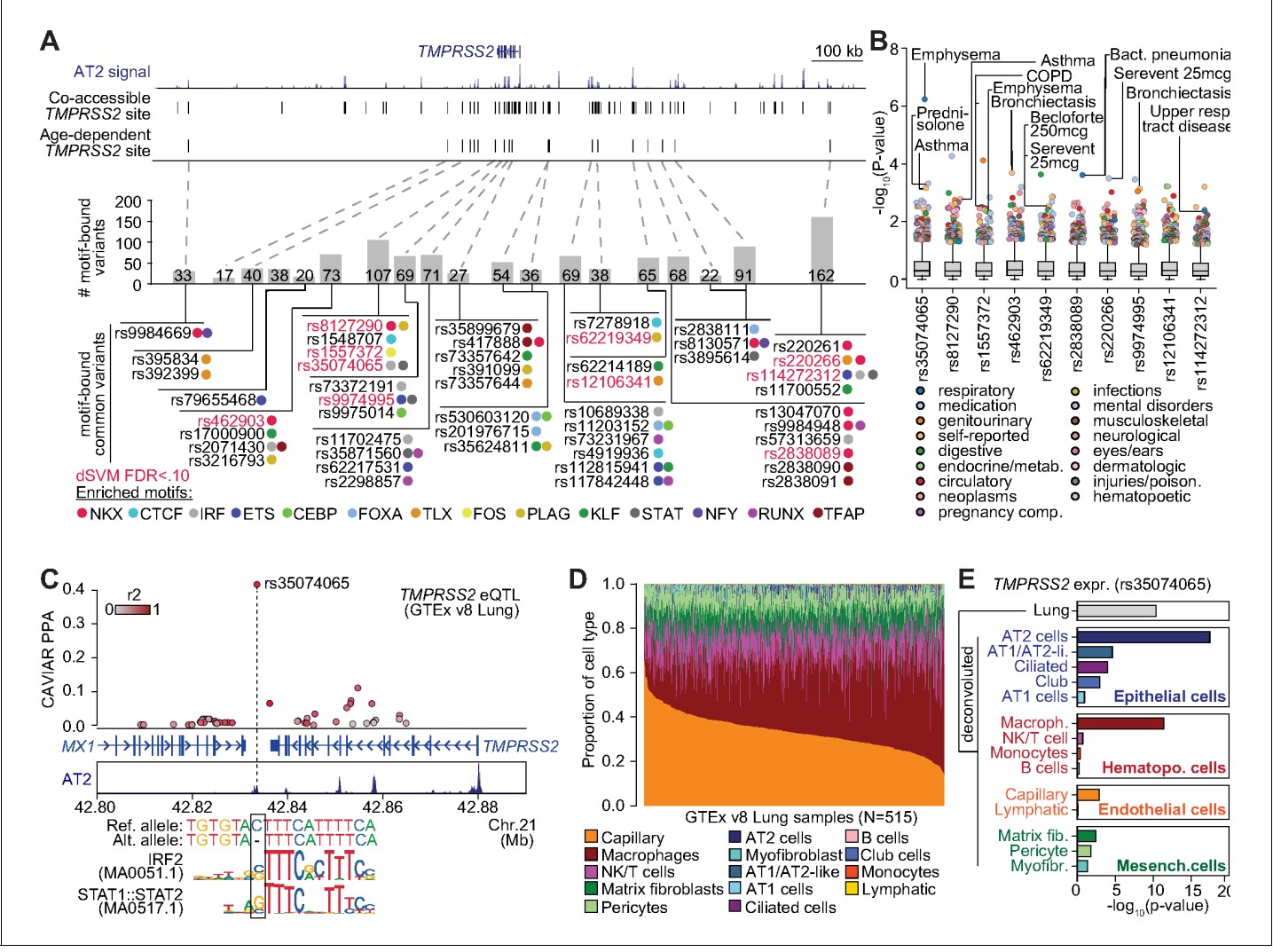

**Figure 4.** Genetic variants predicted to affect age-increasing AT2 accessible chromatin are associated with respiratory phenotypes and *TMPRSS2* expression. (**A**) Top: genome browser view of AT2 sites linked to *TMPRSS2* activity and those with age-dependent increase in accessibility. Middle: Number of non-singleton genetic variants in gnomAD v3 mapping (**Karczewski et al., 2019**) in each age-dependent site predicted to disrupt binding of AT2-enriched TF motifs. Bottom: Common variants (minor allele frequency >0.05 in at least one population) predicted to bind AT2-enriched TF motifs, color-coded by TF family. Motif-bound variants that also have predicted effects (FDR < 0.10) on AT2 accessible chromatin in deltaSVM models (**Lee et al., 2015**) highlighted in red. (**B**) Association of common variants with predicted AT2 effects (motif-disrupting+deltaSVM) with human phenotypes in the UK Biobank (**Lab, 2020**). The majority of tested variants show at least nominal evidence (p<0.005) for association with phenotypes related to respiratory disease, infection and/or medication. (**C**) Fine-mapping probabilities for an *TMPRSS2* expression QTL in human lung samples from the GTEx project release v8 (**GTEx Consortium, 2020**). The variant rs35074065 has the highest casual probability (PPA = 0.42) for the eQTL, maps in an age-dynamic AT2 site and is predicted to disrupt binding of IRF and STAT TFs. Variants are colored based on $r^2$ with rs35074065 in 1000 Genomes Project data using all populations (**Auton et al., 2015**). (**D**) Estimated cell type proportions for 515 human lung samples from GTEx derived using cell-type-specific expression profiles for cell types with more than 500 cells from snRNA-seq data generated in this study. (**E**) Association p-values between rs35074065 genotype and *TMPRSS2* lung expression after including an interaction term between genotype and estimated cell-type proportions for each sample. We observed stronger eQTL association when including an interaction with AT2 cell proportion as well as macrophage proportion.

sequence motifs for IRF and STAT transcription factors, where the *TMPRSS2*-increasing allele disrupted motif binding, suggesting that its effects may be mediated through interferon signaling and anti-viral programs (*Figure 4C*).

As the *TMPRSS2* eQTL at rs35074065 was identified in bulk lung samples, we finally sought to determine the specific cell types driving the effects of this eQTL. Using cell-type-specific gene expression profiles derived from our snRNA-seq data, we estimated the proportions of 14 different cell types present in the 515 bulk lung RNA-seq samples from GTEx v8 (*GTEx Consortium, 2020*; *Figure 4D*). We then tested for association between rs35074065 and *TMPRSS2* expression while including estimated cell-type proportions for each sample in the eQTL model in addition to the covariates used in the original GTEx analysis. We observed highly significant association when including AT2 cell proportion (p=3.8 × 10$^{-18}$) as well as macrophage proportion (p=4.0 × 10$^{-12}$), supporting the possibility that the *TMPRSS2* eQTL at rs35074065 acts through AT2 cells and macrophages, which is in line with *TMPRSS2*-expressing cell types in the lungs (*Figure 4E*, *Figure 2A*, *Figure 2—figure supplement 1B*).

## Fine-mapping risk variants for COVID-19 respiratory failure at the 3p21.31 locus to lung cell-type-specific chromatin sites

Recently the first genome-wide association study of SARS-CoV-2 identified several loci influencing risk of respiratory failure in SARS-CoV-2 infection (*Ellinghaus et al., 2020*). Among these loci, risk variants at the 3p21.31 locus mapped exclusively to non-coding sequences (*Ellinghaus et al., 2020*). We hypothesized that this locus may affect gene regulation in the lungs and used our lung cell-type-specific chromatin accessibility and gene expression map to annotate 3p21.31 risk variants.

Fine-mapping of the 3p21.31 signal resulted in 22 total candidate causal variants. Among these, two fine-mapped variants overlapped a lung cell-type cCRE: rs17713054 (posterior probability [PPA] =0.04), which mapped in a cCRE accessible in epithelial (AT1/2, basal, club, ciliated) and mesenchymal (matrix fibroblast 1/2, myofibroblast) cells with the highest signal in AT2 cells, and rs76374459, (PPA = 0.02), which mapped in a cCRE accessible in erythrocytes (*Figure 5A*). We determined whether these two variants disrupted predicted sequence motifs for relevant TFs. For rs17713054, the minor (and risk increasing) allele A was predicted to bind CEBPA and CEBPB motifs (*Figure 5B*), which were broadly enriched in age-related cCREs in AT2 cells (*Figure 2G*). In further support of CEBP binding to this locus, this variant overlapped a CEBPB ChIP-seq site identified in the ENCODE project (*ENCODE Project Consortium, 2012*; *Wang et al., 2012*; *Figure 5B*). At rs76374459, the risk allele C was predicted to disrupt binding of SPI1 among other TFs and overlapped a SPI1 ChIP-seq site in ENCODE (*ENCODE Project Consortium, 2012*; *Wang et al., 2012*; *Figure 5—figure supplement 1*). Candidate causal variants at the 3p21.31 signal also showed evidence for nominal association with respiratory phenotypes for example bronchiectasis medication (rs76374459 p=2.0×10$^{-3}$), emphysema (rs17713054 p=1.4×10$^{-2}$), and chronic bronchitis (rs17712877 p=1.1×10$^{-2}$), among other associations.

Given multiple fine-mapped variants at 3p21.31 overlapping lung cCREs, we next identified potential target genes of variant activity. We linked sites harboring risk variants to target genes using our single-cell co-accessibility data. The site harboring rs17713054 was co-accessible with the promoter region of multiple genes including *SLC6A20, LIMD1, SACM1L,* and *CCRL2* (*Figure 5C*). Among these genes, *SLC6A20*, which encodes a proline transporter, was expressed predominantly in AT2 cells and had low expression in other cell types (*Figure 5D*). We then asked whether rs17713054 was associated with the expression of linked target genes in the lungs using eQTL data in GTEx v8 (*GTEx Consortium, 2020*). While there were no significant associations, we observed nominal association with *SLC6A20* where the minor (and risk increasing) allele A had increased expression (p=8.09 × 10$^{-3}$). We further tested rs17713054 for association with *SLC6A20* expression including estimated cell-type proportions for each lung sample in the eQTL model (as in *Figure 4E*, see Materials and methods). We observed strongest association when including AT2 or AT1/AT2-like cell proportion (p=4.09 × 10$^{-3}$, p=8.00 × 10$^{-4}$) (*Figure 5E*), supporting the possibility that rs17713054 regulates *SLC6A20* expression in AT2 cells. These results illuminate candidate causal variants mapping in lung cell-type cCREs at the 3p21.31 locus and their putative target genes, which should help guide detailed follow-up study of the mechanism of how this locus contributes to respiratory failure in SARS-CoV-2 infection.

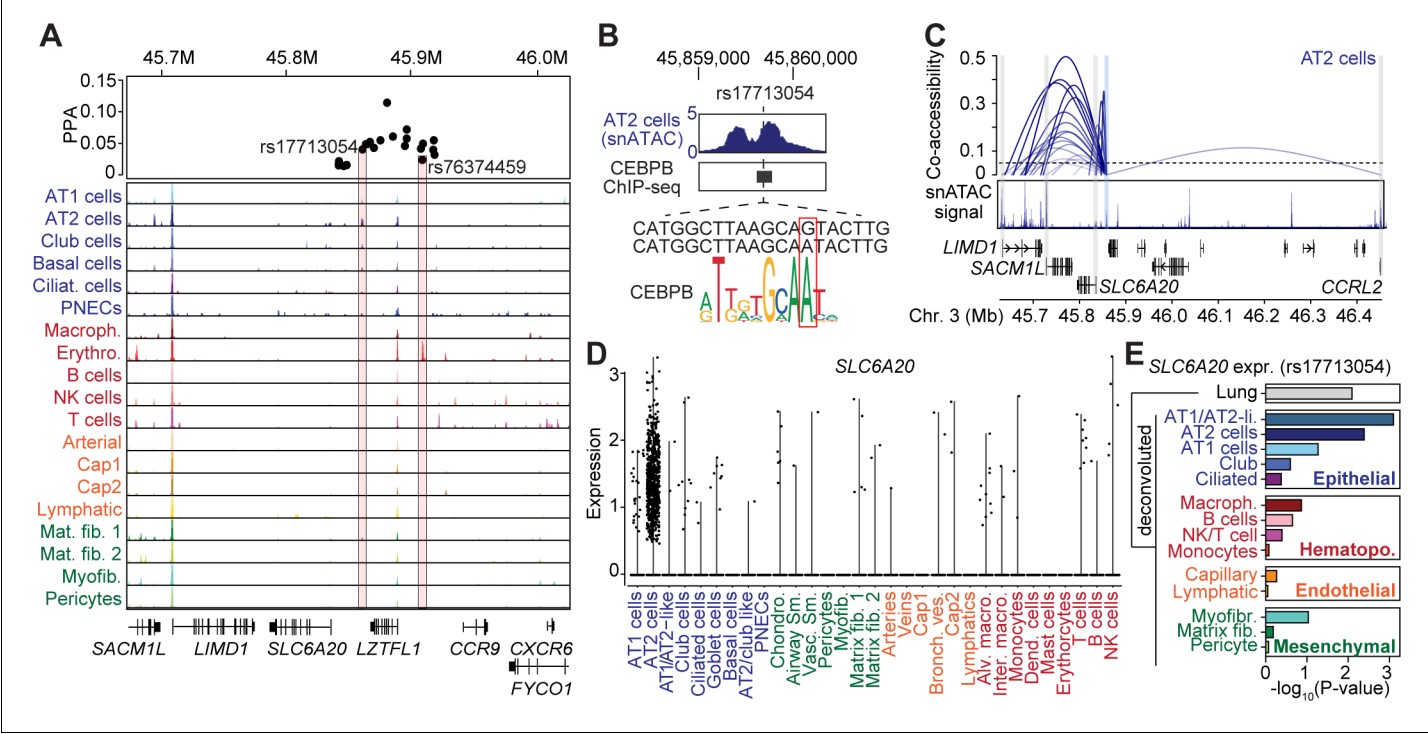

**Figure 5.** Fine-mapped risk variants at the 3p21.31 locus associated with respiratory failure in SARS-CoV-2 overlap lung cell-type chromatin sites. (**A**) Genome browser view (*Robinson et al., 2011*) showing posterior probability (PPA) of variants in the fine-mapping credible set at the 3p21.31 association signal and lung cell-type-specific accessible chromatin profiles. Credible set variants that directly overlap lung cell-type chromatin sites are highlighted. Read depth values represent counts per million (CPM). (**B**) Variant rs17713054 overlaps a site active in AT2 and other epithelial cells and bound by CEBPB among other TFs, and the minor allele A is predicted to bind a CEBP motif. For in-depth analysis of rs76374459 see *Figure 5—figure supplement 1*. Read depth values represent counts per million (CPM). (**C**) Co-accessible links between the site harboring rs17713054 (blue box) and other chromatin sites, including the promoter regions of four genes *SLC6A20, LIMD1, SACM1L* and *CCRL2* (gray box). The height of each link represents the strength of co-accessibility (*Cusanovich et al., 2018*). (**D**) Expression of *SLC6A20* across lung cell types, where each dot represents a nucleus. The highest expression observed was in AT2 cells. (**E**) Association p-values between rs17713054 genotype and *SLC6A20* lung expression after including an interaction term between genotype and estimated cell-type proportions for each sample. We observed strongest eQTL association when including an interaction with AT1/AT2-like and AT2 cell proportion.

The online version of this article includes the following figure supplement(s) for figure 5:

**Figure supplement 1.** Molecular characterization of variant rs76374456.

# Discussion

In this study, we interrogated chromatin accessibility and gene expression in the human lungs at single-cell resolution and identified cCRE predicted to control expression of SARS-CoV2 host entry genes. The lungs came into focus during the COVID-19 pandemic since respiratory failure is a major complication and cause of death (*Du et al., 2020*). Notably, symptoms, severity, and progression of COVID-19 vary considerably between age and population groups (*CDC, 2020a*; *CDC, 2020b*). Our sample-matched snATAC-seq and snRNA-seq datasets from three postnatal stages enabled us to interrogate age-associated dynamics in gene expression and chromatin accessibility. While we focused on COVID-19 related genes in this study, these datasets will more broadly facilitate in-depth analysis of cell-type resolved dynamics of gene regulatory processes in the human lungs.

Using our datasets, we not only corroborated recent findings that the host entry genes *ACE2*, encoding the receptor for the viral spike protein, and *TMPRSS2*, encoding a serine protease for priming of the spike protein, were detected in a higher proportion of AT2 cells in adult lungs

compared to pediatric lungs (*Muus et al., 2020*; *Schuler et al., 2020*), but also identified cCREs linked to *TMPRSS2* and highlighted 19 cCREs with age-increased accessibility. Notably, an increase in accessibility at several of these cCREs predates the onset of gene expression increase, suggesting that, although AT2 cells in childhood stages express lower *TMPRSS2*, the cells may have already acquired the regulatory potential for higher *TMPRSS2* expression. Because these cCREs are predicted to act downstream of immune and inflammatory signals, one plausible implication is that differences in baseline levels of immune/inflammation signaling between children and adults may impact susceptibility to infection by directly regulating the expression of viral entry genes. It is worth noting that these age-related observations are made with the caveat that the sample size of this study is modest (n = 3 individuals per group). Follow-up studies with larger cohorts will be important to reinforce the significance of these findings.

While *ACE2* was detected in a small number of cells and mostly confined to AT2 cells, *TMPRSS2* was expressed in a higher fraction of nuclei predominantly from the epithelial lineage (*Qi et al., 2020*; *Waradon Sungnak et al., 2020*; *Zhao et al., 2020*; *Ziegler et al., 2020*; *Zou et al., 2020*). This may indicate that low ACE2 levels might represent a rate limiting step for viral entry. However, we caution that inhibiting *ACE2* expression may have unintended consequence. Aside from being a viral receptor gene, *ACE2* is also required for protecting the lungs from injury-induced acute respiratory distress phenotypes, the precise cause of COVID-19 mortality (*Imai et al., 2005*). Thus, inhibiting *ACE2* expression may compromise the ability of the lungs to sustain damage. In comparison, *Tmprss2* mutant mice show no defects at baseline and are more resistant to the original SARS-CoV infection (*Iwata-Yoshikawa et al., 2019*; *Kim et al., 2006*). Thus, manipulating the expression of genes such as *TMPRSS2* may represent a safer path to limit SARS-CoV-2 viral entry. *TMPRSS2* is also involved in the entry of other respiratory viruses such as influenza, suggesting that modulating its expression may also be effective in deterring entry and spread of other viruses (*Limburg et al., 2019*).

To explore potential avenues for manipulating the expression of viral entry genes, we identified transcription factors enriched in cCREs with increased chromatin accessibility in adult AT2 cells compared to younger AT2 cells. These included transcription factors involved in stress and immune responses. For example, key interferon pathway-related factors STAT and IRF have binding sites in the age-increased cCREs linked to *TMPRSS2*. The likely causal *TMPRSS2* eQTL variant rs35074065 is predicted to disrupt STAT and IRF binding, raising the possibility that STAT and/or IRF binding at this site may directly control *TMPRSS2* gene expression. Further experimental follow-up studies will be needed to validate the effect of these variants on TF binding and *TMPRSS2* expression, for example using electrophorectic mobility shift assays (EMSA), enhancer/promoter reporter assays, genome editing of in vitro models such as alveolar organoids (*Dobrindt et al., 2020*; *Jacob et al., 2017*). It is interesting that multiple variants linked to *TMPRSS2* were associated with pulmonary function or pulmonary disease medication use. Such association provides plausible links for how pre-existing conditions may modify response to infections.

Finally, and highlighting the utility of our cCRE maps, we reveal a non-coding variant at the 3p21.32 locus risk for COVID-19 related respiratory failure (*Ellinghaus et al., 2020*) overlapping an AT2 cell-active distal cCRE. Importantly this variant (rs17713054) overlaps a binding site for CEBP, a cardinal transcription factor for AT2 cell gene expression (*Xu et al., 2012*). Among the putative target genes for this cCRE was S*LC6A20* which was predominantly expressed in AT2 cells. In *Xenopus* oocytes, *ACE2* expression promotes SLC6A20 protein levels, localization to plasma membrane and its function in proline amino acid transport (*Vuille-dit-Bille et al., 2015*). Conversely, in *Ace2* mutant mice, proline transport, presumably via SLC6A20, was severely disrupted (*Singer et al., 2012*). Further functional studies will be required to validate the molecular effect of this variant on TF binding, enhancer activity and gene regulation in AT2 cells. However, this locus exemplifies how our data provide a foundation to generate testable hypotheses of how risk variants mechanistically contribute to lung disease, in this case that changes in *SLC6A20* expression in AT2 cells may impact severity of SARS-CoV-2 infection of the lungs.

Overall, our study serves as a resource for evolving analyses of gene regulation in the human lungs at cell-type resolution. Moreover, our cCRE maps will also facilitate the interpretation of non-coding genetic variants associated with a broad spectrum of lung diseases including COVID-19 susceptibility and disease severity from emerging GWAS in larger cohorts. We note that this work is a product of the NHLBI-funded LungMap consortium, and our joint goal is to provide the community

with fundamental knowledge of the human lungs to guide the effort to combat COVID-19. We established a web portal to disseminate these datasets to the community: https://www.lungepigenome.org/.

# Materials and methods

**Key resources table**

| Reagent type (species) or resource | Designation | Source or reference | Identifiers | Additional information |
|---|---|---|---|---|
| Peptide, recombinant protein | Tn5 | doi: https://doi.org/10.1101/615179 | | |
| Chemical compound, drug | NEBNext High-Fidelity 2 × PCR Master Mix | NEB | Cat# M0541L | |
| Chemical compound, drug | RNasin Ribonuclease Inhibitor | Promega | Cat# N211B | |
| Chemical compound, drug | DRAQ7 | Cell Signaling | Cat# 7406 | |
| Commercial assay or kit | Chromium Single Cell 3′ Library Construction Kit v3 | 10x Genomics | Cat# 1000075 | |
| Commercial assay or kit | Chromium Single-Cell B Chip Kit | 10x Genomics | Cat# 1000153 | |
| Commercial assay or kit | Chromium i7 Multiplex Kit, 96 rxns | 10x Genomics | Cat# 120262 | |
| Chemical compound, drug | SPRISelect reagent | Beckman Coulter | Cat# B23319 | |
| Software, algorithm | Cell Ranger software package v3.0.2 | 10x Genomics (https://support.10xgenomics.com/single-cell-gene-expression/software/downloads/latest) | | Software |
| Software, algorithm | Seurat v3.1.4 | https://satijalab.org/seurat/ doi:10.1016/j.cell.2019.05.031 | RRID:SCR_016341 | |
| Software, algorithm | DoubletFinder | https://github.com/chris-mcginnis-ucsf/DoubletFinder doi:10.1016/j.cels.2019.03.003 | RRID:SCR_018771 | |
| Software, algorithm | GraphPad Prism version 8.0.0 | www.graphpad.com | RRID:SCR_002798 | |
| Software, algorithm | Trim galore (v.0.4.4) | https://www.bioinformatics.babraham.ac.uk/projects/trim_galore/ | RRID:SCR_011847 | |
| Software, algorithm | BWA (v.0.7.1) | http://bio-bwa.sourceforge.net/ doi:10.1093/bioinformatics/btp324 | RRID:SCR_010910 | Software |

*Continued on next page*

*Continued*

| Reagent type (species) or resource | Designation | Source or reference | Identifiers | Additional information |
|---|---|---|---|---|
| Software, algorithm | Samtools (v. 1.10) | http://www.htslib.org/ doi:10.1093/bioinformatics/btp352 | RRID:SCR_002105 | Software |
| Software, algorithm | Picard | http://broadinstitute.github.io/picard/ | RRID:SCR_006525 | Software |
| Software, algorithm | scanpy (v.1.4.4.post1) | https://github.com/theislab/scanpy | RRID:SCR_018139 | Software |
| Software, algorithm | Harmony (v. 0.1.0) | https://github.com/immunogenomics/harmony doi:10.1038/s41592-019-0619-0 | | Software |
| Software, algorithm | Cicero (v. 1.4.4) | https://github.com/cole-trapnell-lab/cicero-release doi:10.1016/j.molcel.2018.06.044 | | Software |
| Software, algorithm | liftOver | https://genome.ucsc.edu/cgi-bin/hgLiftOver | RRID:SCR_018160 | Software |
| Other | gnomAD v3 | http://gnomad.broadinstitute.org/ doi:10.1038/s41586-020-2308-7 | RRID:SCR_014964 | Database |
| Other | JASPAR 2020 | http://jaspar.genereg.net doi:10.1093/nar/gkz1001 | RRID:SCR_003030 | Database |
| Software, algorithm | FIMO (v. 4.12.0) | http://meme-suite.org/ doi:10.1093/bioinformatics/btr064 | RRID:SCR_001783 | Software |
| Software, algorithm | deltaSVM | http://www.beerlab.org/deltasvm/ doi:10.1038/ng.3331 | | |
| Software, algorithm | MuSiC (v.0.1.1) | https://github.com/xuranw/MuSiC doi:10.1038/s41467-018-08023-x | | |
| Software, algorithm | Python | https://www.python.org/ | RRID:SCR_008394 | |
| Software, algorithm | R (v.3.5.1) | https://www.r-project.org/ | RRID:SCR_001905 | |
| Software, algorithm | Go (v. 1.12.1) | https://golang.org/ | RRID:SCR_017096 | |
| Software, algorithm | NumPy (v.1.16.1) | https://numpy.org/ | RRID:SCR_008633 | python library |
| Software, algorithm | Scikit-learn (v. 0.20.1) | https://scikit-learn.org/stable/ | RRID:SCR_002577 | python library |
| Software, algorithm | seaborn (v. 0.9.0) | https://seaborn.pydata.org/api.html | RRID:SCR_018132 | python library |
| Software, algorithm | MatPlotLib (v.0.9.0) | http://matplotlib.sourceforge.net | RRID:SCR_008624 | python library |
| Software, algorithm | ATACdemultiplex (v. 0.46.12) | https://gitlab.com/Grouumf/ATACdemultiplex/ | | suite of softwares written in GO for snATAC analysis |

*Continued on next page*

*Continued*

| Reagent type (species) or resource | Designation | Source or reference | Identifiers | Additional information |
|---|---|---|---|---|
| Software, algorithm | edgeR (v. 3.22.5) | http://bioconductor.org/packages/release/bioc/html/edgeR.html doi:10.1093/bioinformatics/btp616 | RRID:SCR_012802 | R library |
| Software, algorithm | Matrix (v.1.2–15) | https://cran.r-project.org/web/packages/Matrix/index.html | | R library |
| Software, algorithm | Stringr (v. 1.4.0) | https://www.rdocumentation.org/packages/stringr/versions/1.4.0 | | R library |
| Software, algorithm | Cicero (v. 1.0.14) | https://www.bioconductor.org/packages/release/bioc/html/cicero.html doi:10.1016/j.molcel.2018.06.044 | | R library |
| Software, algorithm | HOMER (v4.11.1) | http://homer.ucsd.edu/homer/download.html doi:10.1016/j.molcel.2010.05.004 | RRID:SCR_010881 | Perl package |
| Software, algorithm | rGREAT (v. 1.20) | https://www.bioconductor.org/packages/release/bioc/html/rGREAT.html for GREAT: doi:10.1038/nbt.1630 | for GREAT: RRID:SCR_005807 | R library |

## Human subjects and tissue collection

Donor lung samples were provided through the federal United Network of Organ Sharing via National Disease Research Interchange (NDRI) and International Institute for Advancement of Medicine (IIAM) and entered into the NHLBI LungMAP Biorepository for Investigations of Diseases of the Lung (BRINDL) at the University of Rochester Medical Center overseen by the IRB as RSRB00047606, as previously described (*Ardini-Poleske et al., 2017*; *Bandyopadhyay et al., 2018*). Portions (0.25–1.0 cm$^3$) of small airway region of right middle lobe (RML) lung tissue were frozen in cryovials over liquid nitrogen and placed at −80°C for storage. Upon request, while kept frozen on dry ice, a tissue piece (approximately 100 mg) was chipped off the sample. These smaller samples were then shipped in cryovials to UCSD on an abundance of dry ice.

## Single-nucleus ATAC-seq data generation

Combinatorial barcoding single-nucleus ATAC-seq was performed as described previously with modifications (*Chiou et al., 2019*; *Fang et al., 2019*; *Cusanovich et al., 2015*; *Preissl et al., 2018*) and using new sets of oligos for tagmentation and PCR (*Supplementary file 8*). Briefly, for each sample, lung tissue was homogenized using mortar and pestle on liquid nitrogen. 1 ml nuclei permeabilization buffer (10 mM Tris-HCL [pH 7.5], 10 mM NaCl, 3 mM MgCl2, 0.1% Tween-20 [Sigma], 0.1% IGEPAL-CA630 [Sigma] and 0.01% Digitonin [Promega] in water; *Corces et al., 2017*) was added to 30 mg of ground lung tissue and tissue was resuspended by pipetting for 8–15 times. Nuclei suspension was incubated for 10 min at 4°C and filtered with 30 μm filter (CellTrics). Nuclei were pelleted with a swinging bucket centrifuge (500 x g, 5 min, 4°C; 5920R, Eppendorf), resuspended in 500 μL high salt tagmentation buffer (36.3 mM Tris-acetate (pH = 7.8), 72.6 mM potassium-acetate, 11 mM Mg-acetate, 17.6% DMF) and counted using a hemocytometer. Concentration was adjusted to 2000 nuclei/9 μL, and 2000 nuclei were dispensed into each well of one 96-well plate. For tagmentation, 1 μL barcoded Tn5 transposomes (*Fang et al., 2019*) was added using a

BenchSmart 96 (Mettler Toledo), mixed five times and incubated for 60 min at 37°C with shaking (500 rpm). To inhibit the Tn5 reaction, 10 µL of 40 mM EDTA were added to each well with a Bench-Smart 96 (Mettler Toledo) and the plate was incubated at 37°C for 15 min with shaking (500 rpm). Next, 20 µL 2 x sort buffer (2% BSA, 2 mM EDTA in PBS) was added using a BenchSmart 96 (Mettler Toledo). All wells were combined into a FACS tube and stained with 3 µM Draq7 (Cell Signaling). Using a SH800 (Sony), 20 2 n nuclei were sorted per well into eight 96-well plates (total of 768 wells) containing 10.5 µL EB (25 pmol) primer i7, 25 pmol primer i5, 200 ng BSA (Sigma). Preparation of sort plates and all downstream pipetting steps were performed on a Biomek i7 Automated Workstation (Beckman Coulter). After addition of 1 µL 0.2% SDS, samples were incubated at 55°C for 7 min with shaking (500 rpm). 1 µL 12.5% Triton-X was added to each well to quench the SDS. Next, 12.5 µL NEBNext High-Fidelity 2 × PCR Master Mix (NEB) were added and samples were PCR-amplified (72°C 5 min, 98°C 30 s, (98°C 10 s, 63°C 30 s, 72°C 60 s)×12 cycles, held at 12°C). After PCR, all wells were combined. Libraries were purified according to the MinElute PCR Purification Kit manual (Qiagen) using a vacuum manifold (QIAvac 24 plus, Qiagen) and size selection was performed with SPRI Beads (Beckmann Coulter, 0.55x and 1.5x). Libraries were purified one more time with SPRI Beads (Beckmann Coulter, 1.5x). Libraries were quantified using a Qubit fluorimeter (Life technologies) and the nucleosomal pattern was verified using a Tapestation (High Sensitivity D1000, Agilent). The library was sequenced on a HiSeq4000 or NextSeq500 sequencer (Illumina) using custom sequencing primers with following read lengths: 50 + 10 + 12 + 50 (Read1 + Index1 + Index2 + Read2). Primer and index sequences are listed in *Supplementary file 8*.

## Single-nucleus RNA-seq data generation

Droplet-based Chromium Single-Cell 3' solution (10x Genomics, v3 chemistry) (*Zheng et al., 2017*) was used to generate snRNA-seq libraries. Briefly, 30 mg pulverized lung tissue was resuspended in 500 µL of nuclei permeabilization buffer (0.1% Triton-X-100 (Sigma-Aldrich, T8787), 1X protease inhibitor, 1 mM DTT, and 0.2 U/µL RNase inhibitor (Promega, N211B), 2% BSA (Sigma-Aldrich, SRE0036) in PBS). Sample was incubated on a rotator for 5 min at 4°C and then centrifuged at 500 rcf for 5 min (4°C, run speed 3/3). Supernatant was removed and pellet was resuspended in 400 µL of sort buffer (1 mM EDTA 0.2 U/µL RNase inhibitor (Promega, N211B), 2% BSA (Sigma-Aldrich, SRE0036) in PBS) and stained with DRAQ7 (1:100; Cell Signaling, 7406). 75,000 nuclei were sorted using a SH800 sorter (Sony) into 50 µL of collection buffer consisting of 1 U/µL RNase inhibitor in 5% BSA; the FACS gating strategy sorted based on particle size and DRAQ7 fluorescence. Sorted nuclei were then centrifuged at 1000 rcf for 15 min (4°C, run speed 3/3) and supernatant was removed. Nuclei were resuspended in 35 µL of reaction buffer (0.2 U/µL RNase inhibitor (Promega, N211B), 2% BSA (Sigma-Aldrich, SRE0036) in PBS) and counted on a hemocytometer. 12,000 nuclei were loaded onto a Chromium Controller (10x Genomics). Libraries were generated using the Chromium Single-Cell 3' Library Construction Kit v3 (10x Genomics, 1000075) with the Chromium Single-Cell B Chip Kit (10x Genomics, 1000153) and the Chromium i7 Multiplex Kit for sample indexing (10x Genomics, 120262) according to manufacturer specifications. CDNA was amplified for 12 PCR cycles. SPRISelect reagent (Beckman Coulter, B23319) was used for size selection and clean-up steps. Final library concentration was assessed by Qubit dsDNA HS Assay Kit (Thermo-Fischer Scientific) and fragment size was checked using Tapestation High Sensitivity D1000 (Agilent) to ensure that fragment sizes were distributed normally about 500 bp. Libraries were sequenced using the NextSeq500 and a HiSeq4000 (Illumina) with these read lengths: 28 + 8 + 91 (Read1 + Index1 + Read2).

## Single-nucleus RNA-seq analysis

Sequencing reads were demultiplexed (cellranger mkfastq) and processed (cellranger count) using the Cell Ranger software package v3.0.2 (10x Genomics). Reads were aligned to the human reference hg38 (Cell Ranger software package v3.0.2). Reads mapping to intronic and exon sequences were retained. Resulting UMI feature-barcode count matrices were loaded into Seurat (*Stuart et al., 2019*). All genes represented in >= 3 nuclei and cells with 500–4000 detected genes were included for downstream processing. UMI counts were log-normalized and scaled by a factor of 10,000 using the NormalizeData function. Top 3000 variable features were identified using the FindVariableFeatures function and finally scaled using the ScaleData function. Barcode collisions were removed for

individual datasets using DoubletFinder (*McGinnis et al., 2019*) with following parameters: pN = 0.15 and pK = 0.005, anticipated collision rate = 10%. Clusters were assigned a doublet score (pANN) and classification as 'doublet' or 'singlet'; called doublets and cells with a pANN score >0 were removed. UMI matrices for datasets were merged and corrected for batch effects due to experiment date, donor, and sex using the Harmony package (*Korsunsky et al., 2019*). UMAP coordinates (*McInnes et al., 2018*) and clustering were performed using the RunUMAP, FindNeighbors, and FindClusters functions in Seurat with principal components 1–23. 25–26, and 28. Clusters were annotated, and putative doublets as defined by expression of canonically mutually exclusive markers were excluded from analysis; remaining cells were re-clustered using the previously described parameters. Final cluster annotation was done using canonical markers. For genes of interest, e.g. *ACE2*, *TMPRSS2*, nuclei with at least one UMI for the gene were considered 'expressing'. To analyze changes in percentage of nuclei expressing we performed two-tailed unpaired t-tests using Graph-Pad Prism version 8.0.0 for Windows, GraphPad Software, San Diego, California USA, www.graph-pad.com.

## Single-nucleus ATAC-seq analysis

For each sequenced snATAC-Seq libraries, we obtained four FASTQ files paired-end DNA reads as well as the combinatorial indexes for i5 (768 different PCR indices) and T7 (96 different tagmentation indices; *Supplementary file 8*). We selected all reads with <= 2 mistakes per individual index (Hamming distance between each pair of indices is 4) and subsequently integrated the full barcode at the beginning of the read name in the FASTQ files (https://gitlab.com/Grouumf/ATACdemultiplex/). Next, we used trim galore (v.0.4.4) to remove adapter sequences from reads prior to read alignment. We aligned reads to the hg19 reference genome using bwa mem (v.0.7.17) (*Li and Durbin, 2009*) and subsequently used Samtools (*Li et al., 2009*) to remove unmapped, low map quality (MAPQ <30), secondary, and mitochondrial reads. We then removed duplicate reads on a per-cell basis using MarkDuplicates (BARCODE_TAG) from the Picard toolkit. As an initial quality cutoff, we set a minimum of 1000 reads (unique, non-mitochondrial) and observed 120,090 cells passing this threshold.

We used a previously described pipeline to identify snATAC-seq clusters (*Chiou et al., 2019*). Briefly, we used scanpy (*Wolf et al., 2018*) to uniform read depth-normalize and log-transform read counts within 5 kb windows. We then identified highly variable (*hv*) windows (min_mean = 0.01, min_disp = 0.25) and regressed out the total read depth across *hv* windows (usable counts) within each experiment. We then merged cells across experiments and extracted the top 50 PCs, using Harmony (*Korsunsky et al., 2019*) to correct for potential confounding factors including donor-of-origin and biological sex. We used Harmony-corrected components to build a nearest neighbor graph (n_neighbors = 30) using the cosine metric, which was used for UMAP visualization (min_dist = 0.3) and Leiden clustering (resolution = 1.5) (*McInnes et al., 2018*; *Traag et al., 2019*).

Prior to the final clustering results, we performed iterative clustering to identify and remove cells mapping to clusters with aberrant quality metrics. First, we removed 3,183 cells mapping in clusters with low read depth. Next, we removed 20,718 cells mapping in clusters with low fraction of reads in peaks. Finally, we re-clustered the cells at high resolution and removed 5,209 cells mapping in potential doublet sub-clusters. On average, these sub-clusters had higher usable counts, promoter usage, and accessibility at more than one marker gene promoter. After removing all of these cells, our final clusters consisted of 90,980 cells. To identify marker genes for each cluster, we used linear regression models with gene accessibility as a function of cluster assignment and usable counts across single cells.

## Computing relative accessibility scores

We define an accessible locus as the minimal genomic region that can be bound and cut by the enzyme. We use $L \subset N$ to represent the set of all accessible loci. We further define a pseudo-locus as the set of accessible loci that relates to each other in a certain meaningful way (for example, nearby loci, loci from different alleles). In this example, pseudo-loci correspond to peaks. We use $\{d_i \mid d_i \subset L\}$ to represent the set of all pseudo-loci. Let $a_l$ be the accessibility of accessible locus $l$, where $l \in L$. We define the accessibility of pseudo-locus $d_i$ as $A_i = \sum_{k \in d_i} a_k$, that is, the sum of accessibility of accessible loci associated with di. Let $C_j$ be the library complexity (the number of distinct

molecules in the library) of cell $j$. Assuming unbiased PCR amplification, then the probability of being sequenced for any fragment in the library is: $s_j = 1 - \left(1 - \frac{1}{C_j}\right)k_j$, where $k_j$ is the total number of reads for cell $j$. If we assume that the probability of a fragment present in the library is proportional to its accessibility and the complexity of the library, then we can deduce that the probability of a given locus $l$ in cell $j$ being sequenced is: $p_{lj} \propto a_l C_j s_j$. For any pseudo-locus $d_i$, the number of reads in $d_i$ for cell $j$ follows the Poisson binomial distribution, and its mean is $m_{ij} = \sum\limits_{k \in d_i} p_{kj} \propto C_j s_j \sum\limits_{k \in d_i} a_k = C_j s_j A_i$. Given a pseudo-locus (or peak) by cell count matrix $O$, we have: $\sum\limits_j O_{ij} = \sum\limits_j m_{ij}$. Therefore, $A_i = Z \dfrac{\sum\limits_j O_{ij}}{\sum\limits_j C_j s_j}$, where $Z$ is a normalization constant. When comparing across different samples the relative accessibility may be desirable as they sum up to a constant, i.e. $\sum\limits_i A_i = 1 \times 10^6$. In this case, we can derive

$$A_i = \frac{\sum\limits_j O_{ij}}{\sum\limits_{ij} O_{ij}} * 10^6.$$

## Calculating the relative percent of cells with accessibility at a locus

To correct for biases occurring from differential read depths between clusters, we used the following strategy to determine the relative ratio of cells with accessibility at a given locus. We defined the set of accessible loci $L$ of a given dataset $D$ as the genomic regions covered by the set peaks $P$ inferred from $D$. We define $X$ the set of cells from $D$, and $S$ a partitioning of $X$. For a given partition $S_i \in S$ and for each feature $p_j \in P$, we computed $m_{ij}$ the ratio of cells from $S_i$ with at least one read overlapping $p_j$. We then defined the score $s_{ij}$ of loci $p_j$ in $S_i$ as $s_{ij} = 10^6 . \dfrac{m_{ij}}{\sum\limits_{j \in P} m_{ij}}$. We finally define the relative ratio of cells normalized across the different clusters as $RS_{ij} = \dfrac{s_{ij}}{\sum\limits_{i \in S} s_{ij}}$.

## Associating promoters to candidate distal regulatory elements

To identify AT2 co-accessible loci linked to the promoters of TMPRSS2, ACE2, FURIN, BSG, and CTSL, we utilized an ensemble approach comprising multiple runs of Cicero analysis. We first performed an independent Cicero analysis for each cluster using a genomic window of 1e6 base pairs. In addition, we enriched these co-accessible links with five runs of cicero analysis using each time a random subset of 15,000 cells from the entire set of nuclei and a genomic window of 250000 base pairs. We then merged the co-accessibility links detected in the five analysis by creating an array of cicero scores for each link. We finally performed a T-test for each link to assess if the average cicero score was different from 0 and filtered links with a p-value<0.10. Secondly, we defined the promoter regions of *TMPRSS2*, *ACE2*, *FURIN*, *BSG*, *CTL*, *CTSL*, and *SLC6A20* as the 1000 bp regions surrounding the TSS gene transcripts related to protein-coding. Finally, we used the pooled list of co-accessible elements to identify all the accessible chromatin sites linked to the promoters.

## Identification and clustering of AT2 peaks with changes in chromatin accessibility genome-wide

We used EdgeR (*Robinson et al., 2010*) to identify differential accessible peaks between each of pair of time points. As input we used the 122,352 peaks in AT2 cell. Dataset ID and sex were used as technical covariates. Sites with False Discovery Rate (FDR) < 0.05 after Benjamini-Hochberg correction were considered significant. Next, we performed K-means using the relative accessibility score with a *loci x timepoints* matrix. We used K from 5 to 8 and computed the Davis-Bouldin index to determine the best K to partition the loci. let $R_{xy} = \frac{(s_x + s_y)}{d_{xy}}$ with $s_x$ the average distance of each sample from cluster $x$ and $d_{xy}$ the distance between the centroids of clusters $x$ and $y$. The Davies-Bouldin index is defined as $DB = \frac{1}{K} \sum\limits_{x,y \in} \max\limits_{x \neq y}(R_{xy})$ and low $DB$ scores indicate better partitioning. We obtained an optimal partition with K=5.

## Identification of AT2 peaks with changes in chromatin accessibility at candidate gene loci

The ensemble of cells $X$ from $D$ can be divided per timepoint, cell subtype, or donor. We identified for individual donors the relative % of cells with at least one read in peaks associated with *ACE2*, *TMPRSS2*, *FURIN*, *BSG*, and *CTSL* promoters. As a background to calculate the relative % of cells, we used the merged set of peaks from all the clusters. Then, we computed a t-test for two independent samples with equal variance for each pair of categories: 30 wk$^{GA}$, 3 yo and 30 yo. For each element the relative % of cells were used as measurement variable and the timepoint as nominal variable.

## Annotation of genomic elements

The GREAT algorithm (*McLean et al., 2010*) was used to annotate distal genomic elements using the following settings: two nearest genes within 1 Mb.

## Transcription factor related analyses

De novo motif enrichment analysis in genomic elements was performed using HOMER (*Heinz et al., 2010*) with standard parameters.

## Predicting variant effects on TF binding and chromatin accessibility

To compile a comprehensive set of variants to test, we downloaded lists of variants from gnomAD v3 (*Karczewski et al., 2019*) and filtered out variants that were singletons or indels longer than 3 bp. We then used the liftOver (*Tyner et al., 2017*) utility to transform GRCh38 into GRCh37/hg19 coordinates, and identified variants overlapping age-dependent AT2 sites linked to *TMPRSS2*. For each variant we obtained sequence surrounding each variant allele and predicted sequence motifs from the JASPAR database (*Fornes et al., 2020*) using FIMO (*Grant et al., 2011*), and focused on motifs of TF families enriched in age-dependent AT2 chromatin. We considered variants with a prediction for at least one allele to have allelic TF binding. We next used deltaSVM (*Lee et al., 2015*) to predict the effects of variants on chromatin accessibility in AT2 cells. First, we extracted the sequences underlying sites co-accessible with the *TMPRSS2* promoter. As described previously (*Chiou et al., 2019*), we trained a sequence-based model of AT2 cell chromatin accessibility and used it to predict effects for all possible combinations of 11mers. We extracted sequences in a 19 bp window around each variant (±9 bp flanking each side). Finally, we calculated deltaSVM z-scores for each variant by predicting deltaSVM scores, randomly permuting 11mer effects and re-predicting deltaSVM scores, and using the parameters of the null distribution to calculate deltaSVM z-scores. From the z-scores, we calculated p-values and q-values and defined variants with significant effects using a threshold of FDR < 0.1. We identified common variants defined as minor allele frequency >0.01 in at least one major population group.

## Phenotype associations for predicted effect variants

We downloaded UK biobank round 2 GWAS combined sex results (*Lab, 2020*; *Sudlow et al., 2015*). We used broad disease categories from the ICD-10-CM to classify ICD10 phenotypes, except for ICD10 codes relating to unclassified symptoms, external causes of morbidity, and factors influencing health status and contact with health services. We combined all non-cancer, self-reported diseases into a single category (self-reported) as well as all treatments and medications (medication). We then extracted GWAS association results for variants that were not tagged as low confidence variants, had significant deltaSVM effects (*Lee et al., 2015*), and mapped in *TMPRSS2*-linked aging-related sites. From these variants, we removed one (rs199938061) which was in perfect linkage disequilibrium with another variant.

## Annotating risk variants at the 3p21.31 locus

We obtained 95% credible sets of fine-mapped variants at the 3p21.31 locus reported in a recent GWAS study of SARS-CoV-2 with severe lung disease (respiratory failure). As variant coordinates were reported in hg38, we manually lifted over variants to hg19 by matching rs IDs to their corresponding genomic coordinates in hg19. We then identified credible set variants overlapping lung cell type chromatin sites. For variants overlapping a site, we obtained sequence surrounding each

variant allele and predicted sequence motifs from the JASPAR database (*Fornes et al., 2020*) using FIMO (*Grant et al., 2011*).

## Deconvoluting lung expression QTLs

We used MuSiC (v.0.1.1) (*Wang et al., 2019*) to estimate the proportions of lung cell types with >500 cells from our scRNA-seq dataset in lung bulk RNA-seq samples from the GTEx v8 release (*GTEx Consortium, 2020*). We combined cell-type labels for capillary (distal and proximal), macrophages (M1 and M2), matrix fibroblasts (1 and 2), and NK/T cells. We modeled the relationship between TMM-normalized *TMPRSS2* or *SLC6A20* expression as a function of the interaction between genotype and cell-type proportion, while considering the covariates used in the original GTEx data including sex, sequencing platform, PCR, five genotype PCs, and 59 inferred PCs from the expression data. From the original inferred PCs, we excluded inferred PC one because it was highly correlated with AT2 cell-type proportion (Spearman $\rho = 0.67$). Including additional covariates in the model such as age, body-mass index or smoking status did not have meaningful impact on the results.

## Statistics

While there was no randomization of samples, and investigators were not blinded to the specimens being investigated, clustering of single nuclei based on transcripts and chromatin accessibility was performed in an unbiased and unsupervised manner, and cell types were assigned after clustering. No statistical methods were used to predetermine sample sizes. To compare fraction of positive cells between samples across ages, a two-tailed unpaired t-test was used. For genome-wide differential accessibility analysis of snATAC-seq peaks, pairwise comparisons between donor age groups (n = 3 per age group) were carried out using EdgeR (*Robinson et al., 2010*) with a cutoff of FDR < 0.05. For locus restricted differential accessibility analysis of snATAC-seq peaks, pairwise comparisons between donor age groups (n = 3 per age group) were made using independent t-test with the same variance assumption. Statistic methods used for other analysis are detailed in the specific method and results sections.

## Code availability

Custom code for processing snATAC-seq datasets is available here: https://github.com/kjgaulton/pipelines/tree/master/lung_snATAC_pipeline; *Wang, 2020*; copy archived at swh:1:rev:2d215946323af71e9d2b158a580c2cf3b41dd5f3.

Custom code used for demultiplexing and downstream analysis for snATAC data is available here: https://gitlab.com/Grouumf/ATACdemultiplex/-/tree/master/ATACdemultiplex, https://gitlab.com/Grouumf/ATACdemultiplex/-/blob/master/scripts/.

## Acknowledgements

We are incredibly grateful to the families who have generously given such precious gifts to support this research. We thank all the members of the LungMAP Consortium for their collaborations. We thank Dr. Bing Ren, Dr. Maike Sander, members of the Sun lab, Gaulton lab, Ren lab and the UCSD Center for Epigenomics for insightful discussions. We thank S Kuan for sequencing and B Li for bioinformatics support. We thank K Jepsen and the UCSD IGM Genomics Center for sequencing the snRNAseq libraries. We thank the QB3 Macrolab at UC Berkeley for purification of the Tn5 transposase.

## Additional information

### Competing interests

Dina A Faddah: employee of and holds stock in Vertex Pharmaceuticals. Kyle J Gaulton: does consulting for Genentech. The other authors declare that no competing interests exist.

## Funding

| Funder | Grant reference number | Author |
|---|---|---|
| National Heart, Lung, and Blood Institute | 1U01HL148867 | Allen Wang<br>Jamie M Verheyden<br>Sebastian Preissl<br>Xin Sun |
| National Heart, Lung, and Blood Institute | U01HL122700 | Gloria Pryhuber |
| HumanTissue Core | U01HL122700 | Gloria Pryhuber |
| HumanTissue Core | HL148861 | Gloria Pryhuber |

The funders had no role in study design, data collection and interpretation, or the decision to submit the work for publication.

## Author contributions

Allen Wang, Kyle J Gaulton, Sebastian Preissl, Conceptualization, Data curation, Formal analysis, Supervision, Funding acquisition, Methodology, Writing - original draft, Project administration, Writing - review and editing; Joshua Chiou, Data curation, Formal analysis, Writing - original draft, Writing - review and editing; Olivier B Poirion, Michael J Valdez, Data curation, Formal analysis, Writing - review and editing; Justin Buchanan, Data curation, Formal analysis, Investigation, Writing - review and editing; Jamie M Verheyden, Validation, Investigation, Writing - review and editing; Xiaomeng Hou, Investigation; Parul Kudtarkar, Dina A Faddah, Data curation; Sharvari Narendra, Visualization; Jacklyn M Newsome, Minzhe Guo, Kai Zhang, Eniko Sajti, Yan Xu, Formal analysis; Randee E Young, Justinn Barr, Validation; Ravi Misra, Heidie Huyck, Lisa Rogers, Cory Poole, NHLBI LungMap Consortium, Resources; Jeffery A Whitsett, Supervision; Gloria Pryhuber, Resources, Funding acquisition; Xin Sun, Conceptualization, Data curation

## Author ORCIDs

Allen Wang (ID) https://orcid.org/0000-0001-9870-7888
Joshua Chiou (ID) http://orcid.org/0000-0002-4618-0647
Jamie M Verheyden (ID) https://orcid.org/0000-0003-4116-8507
Kyle J Gaulton (ID) https://orcid.org/0000-0003-1318-7161
Sebastian Preissl (ID) https://orcid.org/0000-0001-8971-5616
Xin Sun (ID) https://orcid.org/0000-0001-8387-4966

## Decision letter and Author response

Decision letter https://doi.org/10.7554/eLife.62522.sa1
Author response https://doi.org/10.7554/eLife.62522.sa2

# Additional files

## Supplementary files

• Supplementary file 1. Donor metadata tables. Sheet 1: 30wk$^{GA}$ - 30yo: Donor ID, age, sex, race, clinical pathology diagnosis (clinPathDx), gestational age, overall quality of the lung tissue assessment, type of death and cause of death were listed. Not shown are data on body weight, body height, total lung weight and radial alveolar count assessment of alveolarization. All were all within normal limits for age. Abbreviations: DCD: donor after cardiac death; DBD: donor after brain death; GA: gestational age; RDS: respiratory distress syndrome.

• Supplementary file 2. Summary statistics for sequencing libraries.

• Supplementary file 3. Cluster composition and number and fraction of nuclei expressing candidate for SARS-CoV2 cell entry.

• Supplementary file 4. Annotation of peaks co-accessible with candidate genes for SARS-CoV2 cell entry and age-associated changes of chromatin accessibility of peaks co-accessible with *TMPRSS2* promoter.

• Supplementary file 5. GREAT analysis of peaks increasing with age in AT2 cells (groups cIII and cIV in *Figure 3F*).

• Supplementary file 6. De novo motif enrichment analysis of peaks increasing with age in AT2 cells (groups cIII and cIV in *Figure 3F*).

• Supplementary file 7. Genetic variants with predicted functional effects on sites linked to *TMPRSS2*.

• Supplementary file 8. Indexes and primer sequences for snATAC-seq libraries.

• Transparent reporting form

## Data availability

Processed data including the full list of peaks are available for download and can be explored using the web portal https://www.lungepigenome.org. Raw sequencing files has been submitted to Lung-Map Data Collecting Core and will be submitted to dbGAP. Source data for Figure 1—figure supplement 1 is available as Supplementary file 2; Source data for Figure 3B and Figure 3—figure supplement 1A is available as Supplementary file 3. Source data for Figure 3E is available as Supplementary file 4. Source data for Figure 3F is available as Supplementary file 5. Source data for Figure 3G is available as Supplementary file 6. Source data for Figure 4A is available as Supplementary file 7.

The following dataset was generated:

| Author(s) | Year | Dataset title | Dataset URL | Database and Identifier |
|---|---|---|---|---|
| Wang A, Chiou J, Poirion OB, Buchanan J, Valdez MJ, Verheyden JM, Hou X, Guo M, Newsome JM, Kudtarkar P, Faddah DA, Zhang K, Young RE, Barr J, Misra R, Huyck H, Rogers L, Poole C, Whitsett JA, Pryhuber G, Xu Y, Gaulton KJ, Preissl S, Sun X | 2020 | Single Nucleus Multiomic Profiling Reveals Age-Dynamic Regulation of Host Genes Associated with SARS-CoV-2 Infection | https://www.ncbi.nlm.nih.gov/geo/query/acc.cgi?acc=GSE161383 | NCBI Gene Expression Omnibus, GSE161383 |

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
