## [Decision Letter]

**Acceptance summary:**

We believe your study will add important insight into the age related differences in the cellular response and clinical outcomes of SARS-CoV2 disease.

**Decision letter after peer review:**

Thank you for submitting your article "Single Cell Profiling of Human Lung Reveals Cell Type-Specific and Age-Dynamic Control of SARS-CoV2 Host Gene Expression" for consideration by *eLife*. Your article has been reviewed by three peer reviewers, and the evaluation has been overseen by Edward Morrisey as the Senior and Reviewing Editor. The following individual involved in review of your submission has agreed to reveal their identity: Stijn De Langhe (Reviewer #2).

The reviewers have discussed the reviews with one another and the Reviewing/Senior Editor has drafted this decision to help you prepare a revised submission.

1) Please revise your manuscript to acknowledge the limitations of sample size as noted by reviewer #3.

2) Please also revise text to note that further studies will be required to validate the genetic association studies i.e. SNPs as noted by reviewer #1. Also, revise the text to acknowledge the limitations of such SNP association studies and that they may not reveal insight into the functional gene of interest.

3) Please revise the text put your findings in the appropriate context of exacerbating lung disease as noted by reviewer #3.

Reviewer #1:

In this manuscript, Wang et al., provide a large, high quality single cell data set inclusive of snRNA-seq and snATAC-seq from 9 lungs across three stages from premature neonatal life to early adulthood. They then provide an analysis of this data set with a focus on several genes associated with SARS-COV2 infection in the distal lung. They demonstrate that both ACE2 and *TMPRSS2* expression is developmentally regulated, and that *TMPRSS2* is the predominantly detectable SARS receptor in the lung epithelium, expressed on about half of AT2 cells. They then utilize their paired epigenomic data to analyze several putative cis-regulatory elements with changing accessibility with age. Finally, they evaluate these potential regulatory elements in comparison to SNPs previously reported to be associated with increased severity of SARS-COV2 infection. They find that one such SNP is found in a region of *SLC6A20*, providing a plausible mechanism for the association with severe respiratory disease.

In general, this is an important data set and the analysis, while narrow, provides an interesting and timely proof of principle for the use of these data. The observed interactions with age and SNPs are interesting in relationship to SARS-COV2 pathogenesis. The bioinformatic analysis is performed to a high standard and well displayed. In principle, I am in favor of accepting this work for publication in *eLife*.

That said, I think that the SNP, while not needing full biological validation as a true causal allele for SARS-COV2 pathogenesis or severity, should be validated as a functional genetic change. The major thrust of the paper is that the data set can be carefully analyzed with a specific biological question in mind, and can then provide novel and useful insight. I am largely persuaded, but would be entirely convinced if the SNP identified is, in fact, functional with regard to SP1 binding or transcriptional regulatory activity. Relatively simple techniques, including EMSA or luciferase assays, could be reasonably done (even with COVID restrictions) to provide some additional evidence in this regard. I am interested to see what other reviewers feel in this regard. Otherwise, I think the data is of interest to the community and this paper should be published without undue delay.

Reviewer #2:

This is an interesting and timely manuscript Wang et al. investigate the expression and associated cis-regulatory landscape of *ACE2*, *TMPRSS2*, *CTSL*, *BSG*, and *FURIN*, mediators of Sars-Cov-2 infection, at single cell resolution in the non-diseased human lung at 30 days, 3 years and 30 years of age. This is a thorough analysis and the investigators found that the expression of some but not all of these mediators changes according to cell type and age. The authors also identify non-coding variant disrupting binding of CEBP at the 3p21.32 locus risk for COVID-19 related respiratory failure which may regulate *SLC6A20*. Another variant at 3p21.31 disrupted binding of another transcription factor SPI1. Together, these findings will help us understand why certain individual are more prone to Covid19 infection and may aid in finding preventative therapies.

I have no concerns.

Reviewer #3:

This is a very interesting manuscript by Wang et al. that details studies aimed at mapping the transcriptome and chromatin accessibility of lung cells at 30 weeks gestation, 3 years and 30 years postnatally, and using these data to compare expression of genes involved in SARS-CoV2 infection and susceptibility. They use cryobanked tissue samples to perform either single nuclear RNA-Seq or ATAC-Seq. UMAP dimensional reduction of data was used to define cell clusters that are subsequently identified based upon expression of cell type-specific gene signatures. These data were then interrogated for cell type-specific differences in expression and regulation of genes involved in SARS-CoV2 infection, leading to the observation of age-dependent differences in CRE's linked with expression of genes whose products impact viral entry and are predicted to contribute to age-dependent increases in susceptibility. They go on to define genetic variants of SARS-CoV2-related CRE's that are predicted to confer differential regulation of susceptibility genes. Common variants that were identified were linked with increased susceptibility to asthma and emphysema. Some of these variants were validated in silico through analysis of eQTL and ENCODE databases to assess expression levels and candidate transcription factor binding sites that account for altered regulation of variant alleles. These studies are important in that they provide a valuable resource to gain insights into mechanisms contributing to adverse outcomes among patients with COVID-19 and potentially other lung diseases. The application of nuclear RNA-Seq and chromatin accessibility to cryobanked tissue samples collected within the LungMap consortium are truly state-of-the-art and data are of high quality. The only major concern with this study and the value of these data as a resource is the low power that results from analysis of only 3 donor lung tissue samples per age group.

Specific concerns:

1) N = 3 per age group with n = 1 for females and 2 for males in 30 wk GA and 3 yo groups and n=2 for females and 1 for males in the 30 yo group. As such, any statistically relevant analysis requires secondary datasets. Accordingly, statistical analysis in Figure 3 is not valid and the data do not allow analysis of sex as a variable.

2) A minor concern is that there should be discussion of why this approach is of value in defining determinants of human lung disease. This reviewer understands the value of this approach. However, arguably, analysis of genes involved in SARS-CoV2 infection, a pneumotropic virus, is likely to show associations with gene variants linked to emphysema, asthma and bronchiectasis – these respiratory conditions are all associated with viral exacerbations. Discussing this in the context of gene regulation and susceptibility to infection would be of value to the readership.

---

## [Author Response]

1) Please revise your manuscript to acknowledge the limitations of sample size as noted by reviewer #3.

We acknowledged the limitation of the samples size in three places in the manuscript. In the Results for overall snATACseq dynamics: “Given the sample size limitation (n=3 per age group), we acknowledge that the statistical significance of these observed dynamic changes will require further corroboration using datasets from additional donor samples. Nevertheless, we reasoned that because these changes are observable despite modest sample size, the trends provide informative biological insights.” In the Results for *TMPRSS2* dynamics: “Overall, we identified 10 additional cCREs co-accessible with *TMPRSS2* which exhibited patterns of increasing accessibility with age for a total of 19 age-increasing *TMPRSS2*-linked cCREs, 17 of which were statistically significant, with the caveat of modest sample size (N=3 per age group) (FDR < 0.05 via EdgeR and/or p < 0.05 via independent t-test, Figure 3H, I, Figure 3—figure supplement 1C, Supplementary file 4).” In the Discussion, “It is worth noting that these age-related observations are made with the caveat that the sample size of this study is modest (n=3 individuals per group). Follow-up studies with larger cohorts will be important to reinforce the significance of these findings.”

2) Please also revise text to note that further studies will be required to validate the genetic association studies i.e. SNPs as noted by reviewer #1. Also, revise the text to acknowledge the limitations of such SNP association studies and that they may not reveal insight into the functional gene of interest.

We note further follow up studies in the Discussion for *TMPRSS2* follow ups: “Further experimental follow-up studies will be needed to validate the effect of these variants on TF binding and *TMPRSS2* expression, for example using electrophorectic mobility shift assays (EMSA), enhancer/promoter reporter assays, genome editing of in vitro models such as alveolar organoids.” We emphasize the limitation of SNP association studies and how our datasets and future experimental validation are needed for functional insights at the end of the Results: “These results illuminate candidate causal variants mapping in lung cell type cCREs at the 3p21.31 locus and their putative target genes, which should help guide detailed follow-up study of the mechanism of how this locus contributes to respiratory failure in SARS-CoV-2 infection.” And in the Discussion for SNP association with *SLC6A20*: “Further functional studies will be required to validate the molecular effect of this variant on TF binding, enhancer activity and gene regulation in AT2 cells. However, this locus exemplifies how our data provide a foundation to generate testable hypotheses of how risk variants mechanistically contribute to lung disease, in this case that changes in *SLC6A20* expression in AT2 cells may impact severity of SARS-CoV-2 infection of the lung.”

3) Please revise the text put your findings in the appropriate context of exacerbating lung disease as noted by reviewer #3.

To strengthen this important point, we added findings from additional analysis of disease association in the Results, “Candidate causal variants at the 3p21.31 signal also showed evidence for nominal association with respiratory phenotypes for example bronchiectasis medication (rs76374459 P=2.0x10^-3^), emphysema (rs17713054 P=1.4x10^-2^), and chronic bronchitis (rs17712877 P=1.1x10^-2^), among other associations.” In addition, we noted in the Discussion: “It is interesting that multiple variants linked to *TMPRSS2* were associated with pulmonary function or pulmonary disease medication use. Such association provides plausible links for how pre-existing conditions may modify response to infections.”